# DQ-LoRe: Dual Queries with Low Rank Approximation Re-ranking for In-Context Learning

**Jing Xiong**[1][*], **Zixuan Li**[1][*], **Chuanyang Zheng**[2], **Zhijiang Guo**[3][†], **Yichun Yin**[3],
**Enze Xie**[3], **Zhicheng Yang**[4], **Qingxing Cao**[1], **Haiming Wang**[1], **Xiongwei Han**[3]
**Jing Tang**[4,6], **Chengming Li**[7], **Xiaodan Liang**[1,5,8][†]

[1]Sun Yat-Sen University [2]The Chinese University of Hong Kong [3]Huawei Noah's Ark Lab
[4]The Hong Kong University of Science and Technology (Guangzhou) [5]MBZUAI
[6]The Hong Kong University of Science and Technology [7]Shenzhen MSU-BIT University
[8]DarkMatter AI Research

```
{xiongj69, lizx76, caoq, wanghm39}@mail2.sysu.edu.cn,
{cyzheng21}@cse.cuhk.edu.hk,
{guozhijiang, yinyichun, xie.enze, hanxiongwei}@huawei.com
{jingtang}@ust.hk, {licm}@smbu.edu.cn, {yangzhch6, xdliang328}@gmail.com
```

## Abstract

Recent advances in natural language processing, primarily propelled by Large Language Models (LLMs), have showcased their remarkable capabilities grounded in in-context learning. A promising avenue for guiding LLMs in intricate reasoning tasks involves the utilization of intermediate reasoning steps within the Chain-of-Thought (CoT) paradigm. Nevertheless, the central challenge lies in the effective selection of exemplars for facilitating in-context learning. In this study, we introduce a framework that leverages Dual Queries and Low-rank approximation Re-ranking (DQ-LoRe) to automatically select exemplars for in-context learning. Dual Queries first query LLM to obtain LLM-generated knowledge such as CoT, then query the retriever to obtain the final exemplars via both question and the knowledge. Moreover, for the second query, LoRe employs dimensionality reduction techniques to refine exemplar selection, ensuring close alignment with the input question's knowledge. Through extensive experiments, we demonstrate that DQ-LoRe significantly outperforms prior state-of-the-art methods in the automatic selection of exemplars for GPT-4, enhancing performance from 92.5% to 94.2%. Our comprehensive analysis further reveals that DQ-LoRe consistently outperforms retrieval-based approaches in terms of both performance and adaptability, especially in scenarios characterized by distribution shifts. DQ-LoRe pushes the boundary of in-context learning and opens up new avenues for addressing complex reasoning challenges. Our code is released at https://github.com/AI4fun/DQ-LoRe.

## 1 Introduction

Recently, significant advancements in natural language processing (NLP) have been driven by large language models (LLMs) (Chen et al., 2021; Chowdhery et al., 2022; Ouyang et al., 2022; Touvron et al., 2023a;b; Anil et al., 2023; OpenAI, 2023). With the increasing capabilities of LLMs, in-context learning (ICL) has emerged as a new paradigm, where LLMs make predictions based on contexts augmented with a few exemplars (Brown et al., 2020). An important question in the field of in-context learning is how to improve the selection of in-context exemplars to enhance the performance of LLMs (Liu et al., 2022).

---

[*] These authors contributed equally.
[†] Corresponding author

Selecting exemplars for ICL poses challenges due to their instability (Zhao et al., 2021). Even minor changes in the order of samples within exemplars can affect the output (Lu et al., 2022; Su et al., 2023a). The selection of exemplars for LLMs is currently a community-wide trial and error effort, as it is difficult to extract generalizable regularity from empirical observations to form effective selection criteria (Fu et al., 2022; Zhang et al., 2022b). One exception is retrieval-based exemplar acquisition methods (Rubin et al., 2021; Liu et al., 2022; Ye et al., 2023; Li et al., 2023), where a retriever is used to select similar exemplars based on input questions during inference.

However, these methods primarily focus on the similarity between input questions and examples in the training set, without fully exploiting the relationship between intermediate reasoning steps of the given question and other exemplars in the pool. Previous studies have shown that considering such chain-of-thought (CoT) can further improve the performance of LLMs on multi-step reasoning tasks (Wei et al., 2022b; Fu et al., 2022; Gao et al., 2023). Furthermore, some work has also observed that the transfer of knowledge between LLMs and retrievers can effectively enhance the common sense reasoning capabilities of LLMs Xu et al. (2023). Additionally, we observed that prior efforts (Ye et al., 2023; Rubin et al., 2021) struggle to distinguish exemplars in high-dimensional embedding spaces. These observations suggest that exemplar selection based solely on trained question embeddings may suffer from redundant information within the "universal" representations, and may not effectively capture inherent relevance. Removing these redundant information often leads to improved speed and effectiveness (Wang et al., 2023b). The sentence embeddings within the retrieved exemplars often contain similar information, which commonly results in a dense and non-uniform distribution in the vector space. We posit that this is typically due to the embeddings encoding a significant amount of redundant information and exhibiting anisotropy. Employing Principal Component Analysis (PCA; Wold et al. 1987) for dimensionality reduction can assist in filtering out this redundant information and distinguishing between different exemplars, effectively facilitating a more uniform distribution of representations within the vector space.

To address these challenges, we propose a framework that leverages Dual Queries with Low-rank approximation Re-ranking (DQ-LoRe) to incorporate CoTs beyond the input questions, improving the exemplar selection process for in-context learning. DQ-LoRe first queries LLM to generate CoT for a given question. We then concatenate CoT with the question to query the retriever and obtain exemplars from the training pool. We further apply PCA for dimensionality reduction to filter out redundant information and differentiate between different exemplars, improving the selection process.

We conduct extensive experiments on various multi-step reasoning benchmarks to evaluate the performance of DQ-LoRe. The results demonstrate that DQ-LoRe effectively and efficiently selects exemplars, outperforming existing methods. Furthermore, DQ-LoRe exhibits robustness and adaptability in the distribution shift setting, highlighting its versatility across different scenarios. These findings have implications for the use of low-rank constraints in the LLMs paradigm. Our contributions can be summarized as follows:

- We introduce DQ-LoRe, a method that queries supplementary information from LLMs to subsequently re-query a smaller-scale retrieval model. Upon acquiring re-ranked exemplars from the low-rank small model, DQ-LoRe then supplies these exemplars to the LLMs for inference, thereby effectively tackling the challenge associated with the selection of exemplars.
- We employ straightforward and efficient dimensionality reduction techniques to extract crucial reasoning information from the high-dimensional representations of CoTs and questions. This enables the differentiation between various exemplars, particularly distinguishing between exemplars characterized by word co-occurrence and spurious question-related associations and those exemplars that exhibit genuine logical relevance.
- We demonstrate that DQ-LoRe achieves superior performance compared to existing methods and is particularly effective in the distribution shift setting, showcasing its robustness and adaptability across various scenarios.

## 2 RELATED WORK

**In-Context Learning**   LLMs have demonstrated their in-context learning ability with the scaling of model size and corpus size (Brown et al., 2020; Chowdhery et al., 2022; OpenAI, 2023). This ability

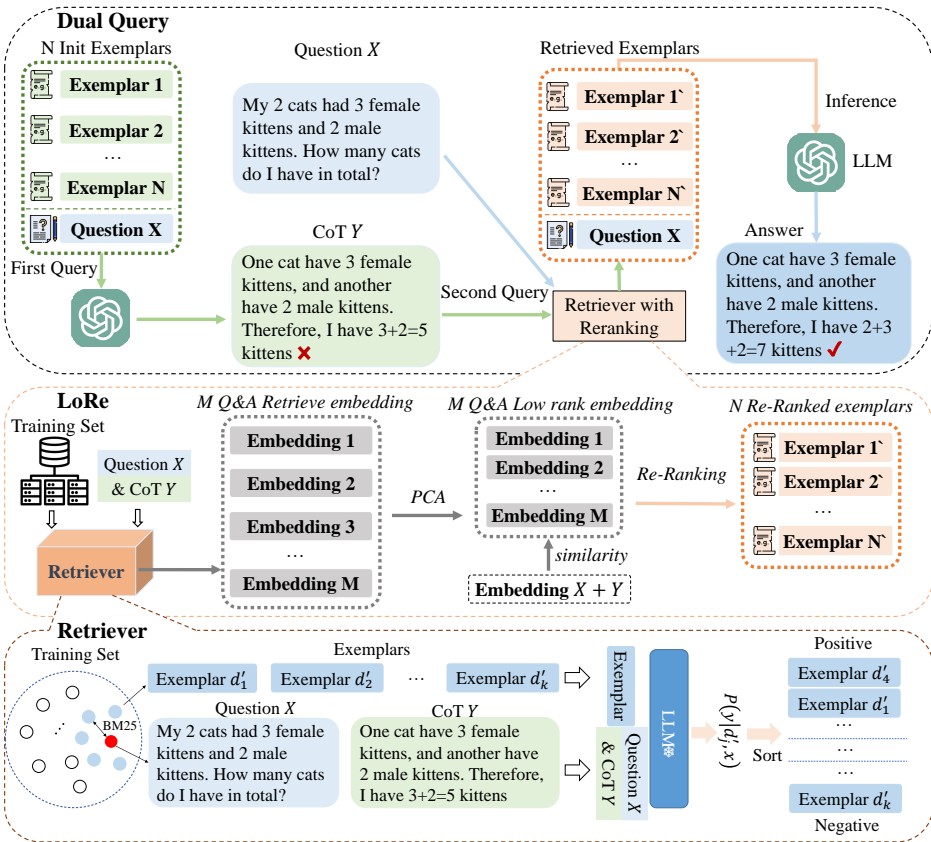

Figure 1: The overall pipeline of DQ-LoRe. It consists of three parts: **Dual Queries** first query LLM to obtain CoT $y$, then query the retriever to obtain the final exemplars via both question and LLM-generated knowledge. **LoRe** leverages PCA to approximate the low-rank embedding of retrieved exemplars, enabling us to better distinguish them. **Retriever** obtains exemplars with similar CoT, through training with positive and negative sets constructed based on CoT similarity produced by BM25 and LLM.

allows language models to learn tasks with only a few exemplars. Several studies have shown that LLMs can successfully perform various complex tasks using in-context learning, including natural language understanding and multi-step reasoning (Shin et al., 2020; Sanh et al., 2022; Liu et al., 2023). In addition to in-context exemplars, Wei et al. (2022b) have explored augmenting the learning process with CoT. CoT involves providing a sequence of intermediate reasoning steps along with the in-context exemplars. Further studies show that the effectiveness of CoT can be enhanced through various approaches. These approaches include breaking down complex questions (Zhou et al., 2022), planning before inference (Wang et al., 2023a), and employing the CoT paradigm for multiple rounds of voting and reasoning (Wang et al., 2022; Zheng et al., 2023). Notably, in the case of multi-step reasoning, in-context learning with CoT has been found to outperform fine-tuning conducted on the same large model with the full training set (Lewkowycz et al., 2022; Wei et al., 2022a).

**Exemplar Selection**   The selection of exemplars for in-context learning is a fundamental question. However, previous studies have highlighted the challenges and instability of exemplar selection (Zhao et al., 2021; Lu et al., 2022; Su et al., 2023a). Even slight changes in the order of samples within exemplars can affect the model's output. The acquisition of exemplars is crucial for enhancing multi-step reasoning capabilities (Liu et al., 2022). Existing efforts mainly focus on the human-designed approach, the vanilla CoT (Wei et al., 2022b) utilizes eight manually written examples, while PAL (Gao et al., 2023) repurposes these exemplars by converting them into programming language statements. Complex-CoT (Fu et al., 2022) selects exemplars with the most complex CoTs from the

training set, resulting in improved performance on multi-step reasoning tasks. Auto-CoT Zhang et al. (2022b) clusters training instances into $k$ categories and selects $k$ samples closest to the cluster center.

Other efforts adopt a retrieval-based method that leverages encoders to encode exemplars and input questions during training (Liu et al., 2022; Rubin et al., 2021; Ye et al., 2023). This enables the selection of exemplars that are close to the vector representation of the input questions. For example, Efficient Prompt Retrieval (EPR; Rubin et al. 2021) models the interaction between input questions and in-context exemplars and optimizes it through a contrastive learning objective to obtain preferred exemplars. Compositional Exemplars for In-context Learning (CEIL; Ye et al. 2023) utilizes Determinantal Point Processes to model the interplay between the provided input and in-context exemplars. This modeling is further enhanced through a meticulously designed contrastive learning objective, with the goal of extracting preferences from language models. Li et al. (2023) proposes a unified retriever to retrieve exemplars for a wide range of tasks. Unlike these methods, we propose to model the relationship between the reasoning process through re-ranking in the representation space after projecting the original representation, enabling better exemplar selection.

## 3 METHODOLOGY

### 3.1 REASONING WITH DUAL QUERIES

As shown in Figure 1, we first query the LLMs to generate CoT, we start with an initial n-shot exemplars. These n-shot exemplars can be retrieved using BM25 based on their semantic similarity to the input question, or other retrieval-based methods such as those proposed in Liu et al. (2022); Rubin et al. (2021); Ye et al. (2023); Zhang et al. (2022b). The exemplars can also include manually designed examples (Wei et al., 2022c; Zhou et al., 2022; Wang et al., 2023a), including CoT and other templates such as Tree-of-Thought (Yao et al., 2023) and Graph-of-Thought (Besta et al., 2023).

In our experiments, we employ the Complex-CoT method (Fu et al., 2022) to obtain the initial n-shot exemplars. This choice is motivated by our observation that using Complex-CoT prompts for querying LLMs can result in CoTs that are richer in inference information. These initial $n$ exemplars and the question $x_i$ are used to query the LLMs and obtain the CoT $y_i$.

With the question $x_i$ and the generated CoT $y_i$, we use the encoder $s_e$ trained in the following section 3.2 to obtain the embedding of the test sample $t_i$, composed of $x_i$ and $y_i$, and all exemplars in the training set.

### 3.2 COT-AWARE RETRIEVER MODEL TRAINING

In this section, we will introduce how to train an encoder to obtain representations of exemplars and test samples. We train a retriever that can measure the similarity between a CoT and a exemplar. Similar to previous studies (Karpukhin et al., 2020; Rubin et al., 2021; Ye et al., 2023), we apply contrastive learning to train a encoder $s_e$ as our retriever. Specifically, we utilize data from the training set to construct training data, where each sample $d_i = (x_i, y_i)$ consists of a question $x_i$ and its corresponding Chain-of-Thought (CoT) denoted as $y_i$, where $i$ refers to the $i$-th data point in the training set.

Given a training sample $d_i$, we construct its corresponding positive and negative set. We first employ BM25 (Robertson et al., 2009) to retrieve the top-k similar training samples as candidate samples from the entire training set, denoted as $D' = \{d'_1, d'_2, ..., d'_k\}$.

After obtaining these $k$ samples, we re-rank them by considering how much the exemplar $d'_j$ close to the $d_i$. We apply a language model (LM) such as text-davinci-003 to calculate the probability:

$$score(d'_j) = P_{LM}(y_i|d'_j, x_i), \quad j = 1, 2, \ldots, k \qquad (1)$$

where $P_{LM}(y_i|d'_j, x_i)$ is the probability of LM generating the CoT $y_i$ given the $d'_j$ and input context $x_i$. Higher $score(d'_j)$ indicates the higher probability of $d'_j$ entails CoT $y_i$ and share the similar reasoning logic. We re-rank the exemplars in $D'$ based on their score. We select the top $t$ samples as positive examples, denoted as $pos_i$, and the last $t$ samples as hard negative examples, denoted as $neg_i$. Typically, $2 * t \leq k$.

During training, we construct the training batch by sampling anchors $d_i$. For each $d_i$, we randomly select one positive $e_i^+$ and one negative example $e_i^-$ from $pos_i$ and $neg_i$. We consider the positive and negative examples of other samples within the same batch as negative for $d_i$. Thus the contrastive loss with $b$ anchors has the following form:

$$Loss\left(x_i, y_i, e_i^+, e_1^+, e_1^-, \ldots e_i^-, \ldots, e_b^-\right) = -\log \frac{e^{\text{sim}\left(x_i, y_i, e_i^+\right)}}{\sum_{j=1}^b e^{\text{sim}\left(x_i, y_i, e_j^+\right)} + \sum_{j=1}^b e^{\text{sim}\left(x_i, y_i, e_j^-\right)}} \tag{2}$$

where sim is the similarity between the anchor sample $d_i = (x_i, y_i)$ and exemplar $d_j$, and is the inner product of their sequence embedding:

$$\text{sim}\left(x_i, y_i, e_i^+\right) = \langle s_e(x_i + y_i), s_e(e_i^+)\rangle. \tag{3}$$

The $s_e$ represents the BERT encoder trained using the aforementioned loss function. After training, we employ $s_e$ as the sentence representation obtained by concatenating the question and CoT. We utilize the trained $s_e$ for retrieving exemplars and compute similarity using vector inner products.

### 3.3 LoRe: Low Rank Approximation Re-ranking

Based on the similarity computed with Equation 3 and select the top-M exemplars $E_M$ to perform the re-ranking. The obtained M exemplars $E_M$ are retrieved based on semantic similarity and often exhibit highly similar CoTs. This results in a mixture of exemplars that exhibit a spurious correlation with the current question and exemplars that are genuinely logically relevant within the CoT, making it difficult to distinguish between them. To address this issue, we employ Principal Component Analysis (PCA) to reduce the embedding dimension of the M exemplars and target sample $t_i$ to the final dimension of $\epsilon$. Subsequently, we recalculate the similarity between each exemplar $e_j$ and $t_i$ with the reduced embedding. For the math reasoning task, we compute the similarity between reduced embeddings using vector inner product. However, for the commonsense reasoning task, in order to distinguish these exemplars while preserving as much CoT information as possible, we employ a Gaussian kernel function to calculate the similarity between embeddings. The Gaussian kernel is expressed as follows:

$$k(s_e(t_i), s_e(e_j)) = \exp\left(-\frac{\|s_e(t_i) - s_e(e_j)\|^2}{2\sigma^2}\right), \tag{4}$$

where $\|s_e(t_i) - s_e(e_j)\|$ denotes the euclidean distance between the represents $s_e(t_i)$ and $s_e(e_j)$. $\sigma$ is a parameter known as the standard deviation of the Gaussian distribution.

Finally, we obtain the top-n exemplars denoted as $E_{N'}$ based on the similarity scores after re-ranking ($M \geq n$). After obtaining $E_{N'}$, we concatenate it with $x_i$ and input it into the LLMs to obtain the final CoT for ICL. With these CoT exemplars, we prompt the LLMs and parse their output to obtain the final answer.

## 4 EXPERIMENT

In our experiments, we evaluate the proposed DQ-LoRe in both independent and identical distribution (*i.i.d.*) and distribution shift settings. In the *i.i.d.* setting, we use the same set of data for training the retriever and exemplar selection during testing. In the distribution shift setting, we train the retriever on one dataset. Then we retrieve exemplars from another dataset during testing. We present the experiment details and results in this section. An introduction to the baselines is provided in the Appendix B. We uniformly conduct our experiments with 8-shot settings.

In constructing the positive and negative samples for training, we set the parameter $k$ to 49 and $t$ to 5. When training the retriever, we use Adam optimizer (Kingma & Ba, 2014)with batch size 16, learning rate le-5, linear scheduling with warm-up and dropout rate 0.1. And we run training for 120 epochs on 8 NVIDIA 3090 GPUs. For each task, we search the LoRe parameter M in $\{16, 32, 64\}$, $\sigma$ in $\{0.25, 0.5\}$ and the LoRe final dimension $\epsilon$ in $\{128, 256, 512\}$.

### 4.1 DATASET

We conduct experiments on three datasets: AQUA (Ling et al., 2017), GSM8K (Cobbe et al., 2021), and SVAMP (Patel et al., 2021). Among these datasets, AQUA and GSM8K have CoT annotation.

Table 1: The accuracy(%) of different models under the *i.i.d.* setting. Complex-CoT selects the most complex CoT from either the annotation or GPT-3.5-Turbo output. All methods select 8-shot exemplars except for CoT, which uses 4-shot manually annotated exemplars. SVAMP* represents the results obtained by training the retriever on the GSM8K dataset and then conducting testing by retrieving exemplars on SVAMP.

| Engine | Model | GSM8K | AQUA | SVAMP | SVAMP* | StrategyQA | QASC |
|---|---|---|---|---|---|---|---|
| | CoT | 55.1 | 35.8 | 77.3 | - | 73.7 | 81.0 |
| | Complex-CoT | 66.8 | 46.5 | 73.0 | 78.3 | 74.0 | 74.1 |
| Text-davinci-003 | Auto-CoT | 60.7 | 42.6 | 80.0 | 81.3 | 70.7 | 73.5 |
| | EPR | 64.6 | 45.0 | **84.6** | 84.6 | 72.9 | 80.2 |
| | CEIL | 63.7 | 47.2 | 75.3 | 81.3 | 72.4 | 80.5 |
| | DQ-LoRe | **69.1** | **48.0** | 83.0 | **85.0** | **74.6** | **82.0** |
| | CoT | 77.0 | 51.9 | 82.0 | - | 73.8 | 81.8 |
| | Complex-CoT | 79.3 | 57.0 | 84.0 | 79.3 | 74.5 | 75.8 |
| GPT-3.5-Turbo | Auto-CoT | 78.4 | 50.4 | 86.0 | 87.3 | 71.2 | 74.1 |
| | EPR | 77.3 | 57.8 | **89.0** | 88.0 | 73.4 | 81.2 |
| | CEIL | 79.4 | 54.7 | 83.7 | 87.3 | 73.4 | 81.8 |
| | DQ-LoRe | **80.7** | **59.8** | 85.3 | **90.0** | **75.4** | **82.7** |

Since the AQUA contains over ten thousand training examples, constructing the positive and negative set for each training data has a high computational cost. Thus, we randomly sample one thousand data from AQUA for training our retriever. In addition to our primary focus on mathematical reasoning datasets, we conducted experiments on commonsense reasoning datasets such as StrategyQA (Geva et al., 2021) and QASC (Khot et al., 2020). Further details can be found in the Appendix C.

It's worth noting that the SVAMP dataset introduces designed perturbations to evaluate whether LLMs learned spurious correlations in math word problems, including question sensitivity, structural invariance, and reasoning ability. Since SVAMP does not have ground-truth CoT annotations, we generate CoT using GPT-3.5-Turbo with Complex-CoT exemplars. For each training data point in these two datasets, we perform eight independent samplings at a temperature of 0.7 and select one correct CoT from the generated results. At last, we acquired 664 training samples with CoTs from SVAMP's training data.

## 4.2 MAIN RESULTS

Table 1 shows the model's performance in the *i.i.d.* setting. It presents that our method achieves the most promising results on the GSM8K and AQUA datasets. On the SVAMP dataset, if the retriever is trained with the generated CoT, our model does not outperform the ERP model. Since the ERP tends to capture and exploit these word co-occurrence patterns. Additionally, in the SVAMP dataset, there is a large number of samples with word co-occurrences between the test and training sets. Therefore, ERP will retrieve all exemplars that have word co-occurrences with the test samples. Our case study in Appendix F presents the same phenomena.

To avoid the impact of these spurious correlations while retrieving on SVAMP and to test the true performance of models, we conduct experiments under conditions of distribution shift. We train the retriever on the GSM8K dataset and conduct retrieval and testing on the SVAMP test set. Under this distribution shift setting, it proves to be effective in neutralizing the influence of spurious correlations, with my model ultimately leading to a commendable 90% accuracy on SVAMP*, significantly surpassing EPR, which suffers from severe spurious correlations. We believe this is due to EPR predominantly relying on word co-occurrence patterns among questions and not considering the similarities between CoTs.

In Table 2, We show the ICL results for GPT-4 on the GSM8K dataset. Our model's performance surpasses previous state-of-the-art retrieved-based method CEIL by a large margin of 1.7% accuracy.

## 4.3 TEST RESULTS FOR DISTRIBUTION SHIFT

We evaluate the robustness of different methods under a distribution shift setting. To create a rigorous evaluation scenario with distribution shift, we introduce the MultiArith (Roy & Roth, 2015) and

Table 2: The accuracy(%) of different ICL methods with GPT-4 on the GSM8K dataset under the *i.i.d.* setting.

| Engine | CoT | Complex-CoT | Auto-CoT | EPR | CEIL | DQ-LoRe |
|--------|-----|-------------|----------|-----|------|---------|
| GPT-4 | 93.0 | 93.4 | 93.1 | 91.3 | 92.5 | **94.2** |

Table 3: The accuracy(%) under the distribution shift setting. Each method is trained on GSM8K and tested on corresponding datasets.

| Engine | Model | SVAMP | MultiArith | SingleEq |
|--------|-------|-------|------------|----------|
| Text-davinci-003 | CoT | 77.3 | 92.3 | **93.8** |
| | Complex-CoT | 78.3 | 91.5 | 93.5 |
| | Auto-CoT | 78.6 | 92.3 | 93.0 |
| | ERP | 75.3 | 92.3 | 92.5 |
| | CEIL | 76.3 | 93.5 | 92.3 |
| | DQ-LoRe | **79.6** | **94.5** | 93.5 |
| GPT-3.5-Turbo | CoT | 82.0 | 98.0 | 95.6 |
| | Complex-CoT | 79.3 | 97.8 | 96.0 |
| | Auto-CoT | 82.6 | 98.0 | 96.0 |
| | EPR | 78.5 | 98.0 | 96.3 |
| | CEIL | 81.2 | 97.3 | 94.8 |
| | DQ-LoRe | **84.0** | **98.5** | **96.5** |

SingleEq Koncel-Kedziorski et al. (2015) datasets, alongside SVAMP. These datasets represent three levels of distribution shift, each posing varying difficulties. shift, each with varying difficulties. Generally, the CoT in the SingleEq dataset are shorter, and we consider it to be the simplest.

We merge the training and testing sample of MultiArith to create a comprehensive test dataset comprising a total of 600 diverse questions. Our goal is to inspect how well an approach adapts to a distinct distribution while relying solely on GSM8K exemplars. This setting can reduce the possibility of high performance bought by the spurious correlation such as co-occurrence patterns among exemplars.

The results are shown in Table 3, our approach exhibits remarkable robustness, particularly on the SVAMP dataset. Our method, which is both trained and retrieved on the GSM8K dataset, successfully reduces the negative effects of word co-occurrence. This underscores the efficacy of our approach in addressing the distribution shift issue and spurious correlation in a variety of contexts.

Moreover, we observe intriguing nuances when examining the performance of our approach on two relatively simple datasets SingleEq and MultiArith. Although careful selection of exemplars yields incremental performance, the simplest configuration of a fixed 8-shot manually designed CoT also achieves competitive performance. In some instances, this straightforward CoT configuration outperforms other methods, particularly on the SingleEq dataset when deployed with the text-davinci-003 engine. These findings emphasize the versatility and potential of our approach across a spectrum of datasets and retrieval scenarios. They also suggest that in situations requiring low-complexity CoTs, meticulous exemplar selection is not effective, and manually designed CoTs can work well.

## 4.4 ABLATION STUDY

In this section, we provide a detailed analysis of the impact of each component on the experimental results. The following results are obtained under the *i.i.d.* setting for GSM8k.

Given that our approach is orthogonal to other retrieval-based methods, we conducted ablation studies on both EPR and CEIL independently. As illustrated in Table 4, "Method + DQ" signifies the implementation of Dual Queries, incorporating both question content and information derived from the CoT. "Method + LoRe" represents the adoption of Low-Rank Approximation Re-ranking, which solely depends on question content, excluding CoT insights. Conversely, "Method + DQ-

Table 4: Ablation Study in EPR and CEIL.

| Method | GPT3.5-Turbo-16k | GPT-4 |
|---|---|---|
| ERP | 77.3 | 91.3 |
| ERP + DQ | 78.3 | 93.0 |
| ERP + LoRe | 77.0 | 90.1 |
| EPR + DQ-LoRe | 80.7 | 94.2 |
| CEIL | 79.4 | 92.5 |
| CEIL + LoRe | 78.7 | 92.0 |
| CEIL + DQ | 79.3 | 92.3 |
| CEIL + DQ-LoRe | 79.9 | 94.1 |

LoRe" indicates the application of the DQ-LoRe approach, integrating both Dual Queries and Low-Rank Approximation techniques for enhanced model performance. This comprehensive evaluation showcases the distinct contributions of each methodological enhancement to the overall efficacy of the models in question.

By comparing the outcomes of "EPR" with "EPR + DQ-LoRe" and "CEIL" with "CEIL + DQ-LoRe", we observed enhancements of 2.9% and 1.6%, respectively, when employing the GPT-4 model. This experimental outcome offers a crucial understanding that the Dual Queries mechanism is essential for the effective operation of LoRe. It implies that executing dimensionality reduction without the supplementary CoT information fails to effectively distinguish among these exemplars. This comparison also underscores the effectiveness of leveraging Quesiton-CoT Pair information, which surpasses the utilization of question information in isolation. These results also demonstrate the versatility of our method, showing that it can be effectively integrated into other approaches.

Table 5: The final accuracy(%) with different initial n-shot exemplars in the *i.i.d.* setting on the SVAMP dataset.

| Engine | Initial Exemplars | SVAMP |
|---|---|---|
| | Random | 78.3 |
| | EPR | 83.6 |
| Text-davinci-003 | CEIL | 81.3 |
| | Scoratic-CoT | **83.7** |
| | Complex-CoT | 83.0 |

## 4.5 THE INFLUENCE OF INITIAL EXEMPLARS

In this section, we analyze the impact of various methods to obtain initial exemplars on the final results. We present the final results in Table 5. It can be observed that the method used to obtain initial exemplars has a significant impact on the final results. Specifically, in our experiment, the term "Random" refers to the random selection of 8 exemplars from the training set during each inference. "EPR" and "CEIL" represent the initial 8-shot exemplars acquired through retrieval on SVAMP. Furthermore, "Scoratic-CoT" involves using decomposed subproblems and solution steps from Complex-CoT exemplars to annotate the SVAMP training set, successfully annotating 624 out of the final 700 training data points with GPT-3.5-Turbo. Subsequently, we conduct training and retrieval using DQ-LoRe with these initial 8-shot exemplars on the resulting Socratic-formatted exemplars. In this experiment, we can discern the impact of different initial prompt formats on the final results. We find that Scoratic-CoT outperforms Complex-CoT under the *i.i.d.* setting on SVAMP, indicating that distinct initial prompt formats have a significant impact on the model's ultimate performance. Moreover, the method of selecting initial exemplars also affects the model's final results. Approaches such as EPR, and CEIL, which carefully select initial exemplars, perform significantly better than a random selection of 8 initial exemplars.

## 4.6 LORE VISUALIZATION

In this section, we provide a comprehensive analysis of the impact of LoRe which employs PCA. We undertake the direct selection of embeddings from the eight exemplars located farthest from our query within the high-dimensional space of the trained encoder. These selected embeddings serve as exemplars during the retrieval process and represent the worst cases. Under the *i.i.d.* setting on the

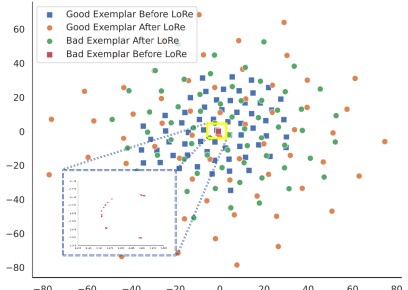 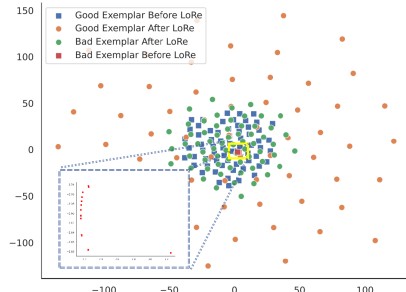

(a) The retriever is trained on the SVAMP dataset and tested on SVAMP, with results presented before and after LoRe.

(b) The retriever is trained on the GSM8K dataset and tested on SVAMP, with results presented before and after LoRe.

Figure 2: T-SNE visualization results of embedding before and after LoRe.

GSM8K dataset, we employ the text-davinci-003 model, resulting in an accuracy of 48.1% using these worst exemplars. This outcome lends credence to the notion that the encoder we have trained possesses the capability to effectively discern between "good" and "bad" exemplars. Building upon this foundation, we proceed to identify and select M exemplars categorized as "good" and "bad" based on the encoder's discernment and visualize the embeddings of M exemplars before and after LoRe dimensionality reduction using t-SNE (Van der Maaten & Hinton, 2012).

On the SVAMP test dataset, the retriever trained on GSM8K achieves better performance than the retriever trained on SVAMP. Hence, we further draw the corresponding t-SNE visualization, which is shown in Figure 2. We initially retrieve 64 embeddings using the retriever in the second query. These embeddings are subsequently subjected to dimensionality reduction. Compared with the "good" and "bad" embeddings of the retriever trained on SVAMP, the "good" and "bad" embeddings of the retriever trained on GSM8K become more distinguished, suggesting that enlarging the difference between the "good" and "bad" embeddings can further improve performance. Figure 2(b) illustrates that before the LoRe PCA process, the distribution of "good" and "bad" embeddings is intermixed. Following the LoRe PCA process, the "good" embeddings migrate outward with a pronounced trend, while the "bad" embeddings exhibit a slight trend in the same direction, leading to an expansion of the gap between them. This divergence contributes to performance improvement. Thus, LoRe's PCA process effectively amplifies the distinction between "good" and "bad" embeddings, further enhancing overall performance. In addition, by comparing Figure 2(a) with Figure 2(b), we can observe that after the LoRe-induced expansion of distances between samples, the dispersion trend of positive samples in Figure 2(b) becomes more pronounced. Conversely, Figure 2(a) shows that after the expansion of distances between samples by LoRe, the gap between positive and negative samples is significantly smaller than the results in Figure 2(b). Another intriguing observation is that before projection, negative samples cluster in a narrow region, whereas positive samples distribute more uniformly across the space. It implies they occupy a narrow conical area in the high-dimensional space. Through LoRe, this conical shape can become more "flattened". The observations indicate that LoRe enhances model performance by modulating the rate of distance diffusion among samples.

## 5 CONCLUSION

In our study, we introduce an innovative approach termed DQ-LoRe, a dual queries framework with low-rank approximation re-ranking that enhances in-context learning for multi-step reasoning tasks. This is achieved by considering the chain-of-thoughts in input questions and exemplars, followed by employing PCA to filter out redundant information in embeddings, and subsequent re-ranking to obtain the final exemplars. This method enhances the model's ability to discern distinctions among various exemplars. Our experimental results demonstrate that DQ-LoRe outperforms existing methods, exhibiting remarkable efficacy, particularly in scenarios involving distribution shifts. This underscores its robustness and versatility across a wide range of situations. We propose that the DQ-LoRe framework will drive progress in LLM retrieval-related research, covering areas such as in-context learning and retrieval-augmented generation.

ACKNOWLEDGMENTS

This work is supported in part by National Key R&D Program of China under Grant No. 2020AAA0109700, Guangdong Outstanding Youth Fund (Grant No. 2021B1515020061), National Natural Science Foundation of China (NSFC)under Grant No.61976233, Mobility Grant Award under Grant No. M-0461, Shenzhen Science and Technology Program (Grant No. GJHZ20220913142600001), Nansha Key RD Program under Grant No.2022ZD014, National Natural Science Foundation of China under Grant No.62006255, CAAl-Huawei MindSpore Open Fund. We thank MindSpore[1] for the partial support of this work, which is a new deep learning computing framwork.

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

# Appendix

## A    Other Related Works

**Low Rank Approximation.**    Principal component analysis (PCA; Wold et al. 1987), is a dimensionality reduction method that is often used to reduce the dimensionality of large data, by transforming a large set of variables into a smaller one that still contains most of the information in the large data. PCA is widely used in various natural language processing tasks (Li et al., 2020; Su et al., 2021; Huang et al., 2021).

Inspired by Li et al. (2020); Su et al. (2021); Huang et al. (2021), after obtaining a set of M exemplars, we perform dimensionality reduction on the embeddings of these M exemplars. We employ PCA Wold et al. (1987), as we believe it can effectively extract key reasoning information from the CoT to differentiate between different exemplars. Furthermore, Devlin et al. (2018) observes that reducing the 768-dimensional embeddings to 256-dimensional vectors from the BERT model could alleviate the issue of anisotropy in the vector space, where vectors are unevenly distributed and confined within a narrow conical space. This finding suggests that unsupervised training of sentence embeddings produces "universal" representations that contain many redundant features for specific domain applications. Removing these redundant features often leads to a dual benefit of enhanced speed and effectiveness (Wang et al., 2023b). Additionally, there are other methods for dimensionality reduction, such as Singular Value Decomposition (SVD) or low-rank approximation techniques (Ye, 2004; Hyvarinen, 1999; Pan et al., 2022). Furthermore, parameter-based training methods for dimensionality reduction have been proposed Hu et al. (2021). In contrast to the methods mentioned above, considering the advantages of not requiring training and having lower computational overhead, we opted for employing PCA for low-rank approximations.

**Spurious Correlation Phenomenon in Math Word Problems.**    In this paper, we primarily investigate how to enhance multi-step reasoning tasks, such as math word problems, through the utilization of improved in-context learning methods. This task has been reported to exhibit a substantial spurious correlation phenomenon in prior research, especially in small-scale language models (Patel et al., 2021; Yang et al., 2022; Xiong et al., 2022). These models struggle to differentiate equivalent solutions, and we observe the corresponding phenomenon as well, where semantic similarity does not necessarily imply logical equivalence in the context of CoT. Despite Math word problems task being nearly solved by LLMs, Shen et al. (2024); Huang et al. (2024) observe that recent models still perform significantly below elementary students. Previous works (Li et al., 2021; Zhang et al., 2022a) have employed contrastive learning to mitigate this bias by aligning the model's intermediate representations when producing equivalent solutions. Recent work employs data distillation (Yu et al., 2023; Lu et al., 2024; 2023) and meticulous assembly of single-step reasoning (Su et al., 2023b) to mitigate bias in LLMs duringå reasoning tasks. Inspired by Karpukhin et al. (2020), in this paper, we also employ contrastive learning techniques to train the associativity between our CoTs and exemplars, enhancing the robustness of the encoder.

## B    Introduction of Our Baselines

We compare DQ-LoRe to an extensive set of baselines and state-of-art models, details are provided as follow:

**CoT** (Wei et al., 2022b) has delved into enhancing the learning process through the incorporation of CoTs, which entails presenting a series of intermediate reasoning steps in conjunction with the relevant in-context example. CoT can be applied across various domains, including mathematical, logical, symbolic, and any area where complex reasoning is required.

**Complex-CoT** (Fu et al., 2022) enhances CoT on complex reasoning tasks by selecting exemplars with the most complex CoTs from the training set. Specifically, it samples multiple reasoning chains from the model and then chooses the majority of generated answers from complex reasoning chains over simple ones. When used to prompt LLMs, Complex-CoT substantially improves multi-step reasoning accuracy.

**Auto-CoT** (Zhang et al., 2022b), on the other hand, leverages diversity within the training set to identify CoTs with maximal distinctions. The rationale behind this approach is to extract a richer spectrum of information. Empirical findings from their experiments substantiate this perspective.

**EPR** (Rubin et al., 2021) aims to retrieve prompts for in-context learning using annotated data and a language model. Given an input-output pair, EPR estimates the probability of the output given the input and a candidate training example as the prompt, and label training examples as positive or negative based on this probability. Next, an efficient dense retriever is trained from this data, which is used to retrieve training examples as prompts at test time. During this process, the interaction between input questions and in-context examples is better modeled. Optimizing this interaction through a contrastive learning objective helps to identify and prioritize preferred exemplars.

**CEIL** (Ye et al., 2023) formulates in-context example selection as a subset selection problem. Specifically, CEIL employs determinantal point process to capture the interaction between the provided input and in-context examples. It is refined through a meticulously crafted contrastive learning objective, aiming to consider both the relevance of exemplars to the test questions and the diversity among the exemplars.

## C  EXPERIMENTS ON COMMONSENSE REASONING DATASETS

In this section, we present experiments on commonsense reasoning datasets such as StrategyQA (Geva et al., 2021) and QASC (Khot et al., 2020) using GPT-3.5-Turbo-16k. The detailed experimental results are presented in the table below:

Table 6: Experiments on Commonsense Reasoning Datasets.

| Model | StrategyQA | QASC |
|---|---|---|
| CoT | 73.8 | 81.8 |
| Complex-CoT | 74.5 | 75.8 |
| Auto-CoT | 71.2 | 74.1 |
| EPR | 73.4 | 80.2 |
| CEIL | 73.4 | 80.8 |
| DQ-LoRe(PCA) | 74.6 | 81.2 |
| DQ-LoRe(Gaussian kernel) | **75.4** | **82.7** |

For the commonsense reasoning dataset, we introduce the LoRe model utilizing a Gaussian kernel Scholkopf et al. (1997), wherein the conventional PCA step is supplanted by employing a Gaussian kernel to ascertain the similarity between the embedding of the exemplar and that of the current inference query combined with the CoTs embedding (derived from the first query). This modification ensures a nuanced preservation and differentiation of information within the embeddings, facilitating a more refined and contextually relevant analysis for complex commonsense reasoning tasks.

The motivation for adopting the kernel method is to preserve as much information as possible in the embeddings while also distinguishing these embeddings in the commonsense reasoning task. Although this is not a dimensionality reduction approach, the fundamental idea remains unchanged. First, the embeddings of **M** similar exemplars are queried, then mapped into another vector space. After this mapping, the similarity of these embeddings to the current problem's question+CoT embedding is recalculated, followed by re-ranking to yield the final **N** exemplars (**M>N**).

In the realm of commonsense datasets, our approach remains state-of-the-art, and an interesting discovery is that using a kernel trick to map the exemplar's embedding into a higher-dimensional space for re-ranking, instead of PCA for dimensionality reduction, can effectively enhance the model's performance. There is even a 1.5% performance improvement on the QASC dataset compared to PCA. This is because commonsense reasoning tasks have a significant difference compared to solving math word problems. Specifically, the background knowledge required for math word problems is presented within the question itself, while the factual information for commonsense reasoning needs to be queried from LLMs. Moreover, this task is sensitive to textual details. Therefore, filtering information in embeddings is not a suitable approach for commonsense reasoning, as dimensionality reduction might eliminate some crucial entity information. This leads to difficulties in distinguishing

their CoT in the embedding space. Hence, we have adopted a kernel method to transform the embeddings in commonsense reasoning tasks, ensuring that the information within the embeddings is preserved while still becoming separable.

## D    EXPERIMENTS ON MORE LLMS

In this section, we present experiments on a broader set of LLMs using the GSM8K dataset. The results are presented in the table below. Due to computational resource constraints, our experiments are primarily conducted on three 7 billion-parameter scale models: Llama2-7b-hf Touvron et al. (2023c), Llama2-7b-chat-hf Touvron et al. (2023c), and Vicuna-7b Chiang et al. (2023).

Table 7: Experiments on 7B-Scale Language Models on the GSM8K Dataset.

| Model | Llama2-7b-hf | Llama2-7b-chat-hf | Vicuna-7b |
|---|---|---|---|
| CoT | 12.6 | 22.9 | 19.4 |
| Complex-CoT | **17.7** | **29.4** | 23.8 |
| Auto-CoT | 15.0 | 27.0 | **23.9** |
| EPR | 15.1 | 23.0 | 22.0 |
| CEIL | 15.2 | 26.7 | 22.4 |
| DQ-LoRe | 16.0 | 28.9 | 23.8 |

We observe that compared to the retrieval baselines (EPR and CEIL), our method demonstrates significant improvements even on models with smaller open-source parameter sizes. Moreover, the effectiveness of our approach increases with the improvement in the model's instruction-following capability. However, we also acknowledge that, when compared to Complex-CoT and Auto-CoT, our method does not show as significant improvements on the LLaMa family 7 billion scale models.

## E    ABLATION STUDY

Table 8: Experiments with Different Encoders and Ablation of LoRe Across Baselines on GSM8K.

| Method | BERT-base | RoBERTa-base |
|---|---|---|
| EPR | 77.3 | 78.0 |
| EPR + LoRe | 77.0 | 77.8 |
| CEIL | 79.4 | 77.7 |
| CEIL + LoRe | 78.7 | 77.4 |
| DQ-LoRe w/o LoRe | 78.9 | 76.4 |
| DQ-LoRe | **80.7** | **78.8** |

**Analysis of the Impact of Different Encoder.**    To investigate the impact of different encoder models on retrieval performance, we examine the effects of employing various base models on the experimental results using GPT-3.5-Turbo-16k. Our findings indicate that, in terms of overall performance, the "BERT-base" model outperforms the "RoBERTa-base" (Liu et al., 2019) model, except for an enhancement observed in EPR when employing the "RoBERTa-base" model. Furthermore, we assess the influence of LoRe on various models and note that the adoption of LoRe leads to a decline in performance for both EPR and CEIL. This decline can primarily be attributed to the lack of the dual queries process, which impedes the incorporation of additional CoT information. As demonstrated in Section 4.4, conducting LoRe without directly encoding additional CoT information leads to diminished in-context learning performance.

**Analysis of the Impact of Regularization Method.**    To present a comprehensive analysis of the impact of different regularization methods on the embedding space, we have executed additional experiments across various datasets. In the table 9, the "w/o dual queries" denotes the scenario where re-ranking is performed without engaging the query language model, a practice that has

Table 9: Ablation Study: Impact of Various Regularization Techniques in Re-Ranking Algorithm

| Re-ranking method | GSM8K | SVAMP | AQUA | StrategyQA | QASC |
|---|---|---|---|---|---|
| w/o dual queries | 77.0 | 86.1 | 54.9 | 72.8 | 80.1 |
| w/o re-ranking | 78.9 | 89.0 | 57.0 | 74.1 | 82.1 |
| Gaussian kernel | 79.0 | 87.6 | 56.2 | **75.4** | **82.7** |
| MGS | 79.7 | 87.3 | 57.8 | 73.2 | 81.0 |
| ICA | 79.5 | 85.7 | 57.0 | 74.2 | 82.2 |
| PCA | **80.7** | **90.0** | **59.8** | 74.6 | 81.2 |
| BERT-base | **80.7** | **90.0** | **59.8** | **75.4** | **82.7** |
| RoBERTa-base | 78.8 | 89.0 | 57.8 | 74.2 | 81.2 |

been empirically shown to detrimentally affect model performance. It highlights the significance of incorporating dual queries mechanisms to enhance the model's ability to discern and prioritize CoT information effectively. The "w/o re-ranking" denotes scenario where no re-ranking process is implemented. It implies that the algorithm conducts a singular sorting operation following the query to LLMs. The term "ICA" (Independent Component Analysis) (Hyvärinen & Oja, 2000) represents a theoretically sound approach to achieving disentangled representations, which underscores the capacity to discern latent variables within the data. In this method, we reduce the feature dimensions of the embedding to 16 dimensions. In the table 9, "BERT-base" and "RoBERTa-base" are denoted as employing distinct encoders for initial-stage ranking similarity computations, and it's noteworthy that they utilize the Gaussian kernel as the regularization method specifically for the StrategyQA and QASC datasets.

From the results obtained, it is evident that the selection of regularization techniques significantly influences the re-ranking algorithm's efficacy. Specifically, in the task of Math Word Problems, the employment of PCA and MGS methods is associated with an enhancement in performance. This indicates that the efficacy of the re-ranking process can be attributed not only to the dimensionality reduction of vectors but also to their orthogonalization, with both factors playing pivotal roles. Conversely, in commonsense reasoning tasks, the Gaussian kernel approach yields a notable enhancement in outcomes. It is important to recognize the distinct challenges posed by commonsense reasoning task in comparison to math reasoning task. The requisite background knowledge for Math Word Problems is inherently contained within the questions themselves, whereas commonsense reasoning tasks demand the retrieval of factual information from LLMs. Moreover, commonsense reasoning tasks exhibit a heightened sensitivity to textual nuances.

Consequently, the application of information filtering within embeddings is not recommended for commonsense reasoning tasks due to the potential loss of critical entity information through dimensionality reduction. To address this, a Gaussian kernel has been employed to modify the embeddings, ensuring the preservation of information within the embeddings while still achieving separability.

Based on the aforementioned observations, we can draw a conclusion that different regularization techniques are required for re-ranking across different tasks. In future work, we aim to identify a unified regularization technique to handle all tasks, which may require further integration of kernel method and PCA.

Table 10: Comparing the Using of Trained Encoder and Solely CoT Similarity Retrieval.

| Method | w/o trainning | with trainning |
|---|---|---|
| question + CoT | 77.7 | 78.9 |
| question + CoT with LoRe | **78.8** | **80.7** |
| only CoT | 76.1 | 79.4 |
| only CoT with LoRe | 78.1 | 80.4 |

**Analysis of the Impact of Trained Encoder.** In Table 10, we explore the impact of various encoder training strategies and ranking methodologies on the outcome. The "w/o training" configuration

utilizes a BERT model, which has not been trained on a specific task, for encoding sample representations. Conversely, the "with training" setup involves a BERT model fine-tuned on a training dataset, aimed at enhancing encoding effectiveness through contrastive learning. The "question + CoT" scenario leverages both the query context and CoT insights simultaneously for encoding, while the "only CoT" approach focuses solely on encoding based on CoT information, establishing direct similarity measures between the CoT in the exemplars and the question's CoT.

A comparative analysis between the "w/o training" and "with training" conditions reveals that encoder training on pertinent datasets markedly improves retrieval performance. Especially notable in the "only CoT" setting, training the encoder results in a 3.3% performance improvement. Our proposed method demonstrates resilience, consistently enhancing outcomes even without dataset-specific encoder training. Remarkably, in the "only CoT with LoRe" configuration under "w/o training", we attain a 2% absolute performance boost, underscoring the efficacy of our approach without necessitating specialized encoder training. Lastly, our findings suggest that relying solely on CoT similarity for encoding is less effective compared to a hybrid approach that incorporates both question and CoT, indicating the added value of integrating comprehensive contextual cues in the encoding process.

Table 11: Average Processing Time per Question for DQ-LoRe and Baseline.

| Stage | EPR | CEIL | DQ-LoRe |
|---|---|---|---|
| All Time | 0.756s | 0.758s | 1.584s |
| First Query | - | - | 0.728s |
| Second Query | 0.029 | 0,032s | 0.028s |
| Re-Ranking | - | - | 0.101s |
| Inference | 0.727s | 0.726s | 0.727s |

**Analysis of Time Consumption.** We conduct thorough tests to compare the time efficiency of our approach against baseline methods. Subsequently, we conducted a detailed analysis of the time consumption for each specific module within our method. For a more comprehensive understanding, please refer to the detailed experimental results provided in the subsequent table.

From Table 11, it is evident that the time expended by our method exhibits a linear relationship in comparison to the baseline. Further insights from Table 11 reveal that the predominant additional consumption, in contrast to the baseline, occurs during the initial stage of requesting the LLMs to acquire CoTs. In practice, each API key for the GPT-3.5-Turbo-16k model can process 180,000 tokens per minute, and with an average request requiring around 2000 tokens, theoretically, by employing numerous keys and parallelizing threads, the time consumption in this stage can be significantly reduced.

## F    CASE STUDY ON SVAMP

In this section, we present exemplars retrieved for a single data point on the SVAMP dataset. Tables 12, 13, 14, 15, and 16 respectively display the 8-shot results retrieved by EPR, CEIL, and DQ-LoRe. It can be observed that EPR relies on the pattern of word co-occurrence for retrieval, while CEIL retrieves some unrelated examples. From Tables 12, 13, it is visually evident that although EPR achieves the highest accuracy on SVAMP, it relies on patterns of word co-occurrence and happens to find the right prompt as a shortcut. This observation reveals that EPR has learned the phenomenon of spurious correlations. The model exhibits a tendency to group together all exemplars featuring word co-occurrence, which is prevalent in the SVAMP dataset due to its numerous analogous questions that mainly differ in numerical values. Consequently, EPR consistently retrieves exemplars characterized by significant word co-occurrence and analogous problem-solving methodologies.

However, our model not only identifies exemplars with word co-occurrence and similar problem-solving approaches but, more importantly, it can also find exemplars that lack word co-occurrence but share a common problem-solving approach. This implies that our model can discover deeper logical associations and connections between exemplars.

Table 12: EPR Case Study on SVAMP: Part 1

**Problem**

Question: Mary is baking a cake. The recipe calls for 6 cups of flour 8 cups of sugar and 7 cups of salt. She already put in 5 cups of flour.How many more cups of sugar than cups of salt does she need to add now?

**EPR Prompt**

Question: Mary is baking a cake. The recipe calls for 12 cups of flour 14 cups of sugar and 7 cups of salt. She already put in 2 cups of flour.How many more cups of flour than cups of salt does she need to add now?

A: Let's think step by step To find out how many more cups of flour than cups of salt Mary needs to add now, we first need to calculate the total amount of flour and salt needed for the recipe. The recipe calls for 12 cups of flour and Mary has already added 2 cups, so she still needs 12 - 2 = 10 cups of flour. The recipe also calls for 7 cups of salt and Mary has not added any yet, so she needs 7 cups of salt. Therefore, Mary needs to add 10 - 7 = 3 more cups of flour than cups of salt. The answer is 3.

Question: Mary is baking a cake. The recipe calls for 9 cups of sugar 7 cups of flour and 4 cups of salt. She already put in 2 cups of flour.How many more cups of flour than cups of salt does she need to add now?

A: Let's think step by step Mary needs to add 7 - 2 = 5 more cups of flour. She needs to add 5 - 4 = 1 more cup of flour than cups of salt. Answer: $\boxed{1}$ . The answer is 1.

Question: Mary is baking a cake. The recipe calls for 14 cups of flour and 6 cups of sugar. She already put in 5 cups of flour.How many more cups of flour than cups of sugar does she need to add now?

A: Let's think step by step Mary needs to add 14 - 5 = 9 more cups of flour. She needs to add 9 - 6 = 3 more cups of flour than cups of sugar. Answer: $\boxed{3}$ . The answer is 3.

Question: Mary is baking a cake. The recipe calls for 11 cups of flour and 7 cups of sugar. She already put in some cups of flour. If she still needs 2 more cups of flour than sugarHow many cups of flour did she put in?

A: Let's think step by step. Let's assume Mary already put x cups of flour. According to the information given, she still needs 2 more cups of flour than sugar, so she needs (7 + 2) = 9 cups of sugar. Since the recipe calls for 11 cups of flour, the total amount of flour needed is 11 cups. Therefore, the equation is x + 9 = 11. Subtracting 9 from both sides of the equation, we get x = 11 - 9 = 2. Mary has already put in 2 cups of flour. The answer is 2.

Question: Mary is baking a cake. The recipe calls for 11 cups of sugar and some more cups of flour. She already put in 3 cups of flour. If she still needs to add 6 more cups of flourHow many cups of flour did the recipe require?

A: Let's think step by step: Mary already put in 3 cups of flour, and she still needs to add 6 more cups of flour. So, the total amount of flour she needs to add is 3 + 6 = 9 cups. Therefore, the recipe requires 9 cups of flour. The answer is 9.

Question: Mary is baking a cake. The recipe calls for 3 cups of sugar 10 cups of flour and 15 cups of salt. She already put in 6 cups of flour.How many more cups of flour does she need to add?

A: Let's think step by step To find out how many more cups of flour Mary needs to add, we need to subtract the amount of flour she already put in from the total amount of flour required. The total amount of flour required is 10 cups, and Mary already put in 6 cups. Therefore, she needs to add 10 - 6 = 4 more cups of flour. Mary needs to add 4 cups of flour. The answer is 4.

Table 13: EPR Case Study on SVAMP: Part 2

**Problem** Question: Mary is baking a cake. The recipe calls for 9 cups of flour and 11 cups of sugar. She already put in 4 cups of flour.How many more cups of sugar than cups of flour does she need to add now?

A: Let's think step by step Mary needs to add 9 - 4 = 5 more cups of flour. Mary needs to add 11 - 5 = 6 more cups of sugar than cups of flour. Answer: $\boxed{6}$. The answer is 6.

Question: Mary is baking a cake. The recipe calls for 12 cups of sugar and 14 cups of flour. She already put in 10 cups of sugar.How many more cups of flour than cups of sugar does she need to add now?

A: Let's think step by step The recipe calls for 12 cups of sugar and she already put in 10 cups, so she needs to add 12 - 10 = 2 more cups of sugar. The recipe calls for 14 cups of flour and she hasn't added any yet, so she needs to add 14 cups of flour. To find the difference between the cups of flour and cups of sugar, we subtract the cups of sugar from the cups of flour: 14 - 2 = 12. She needs to add 12 more cups of flour than cups of sugar. The answer is 12.

Table 14: CEIL Case Study on SVAMP

**Problem**

Question: Mary is baking a cake. The recipe calls for 6 cups of flour 8 cups of sugar and 7 cups of salt. She already put in 5 cups of flour.How many more cups of sugar than cups of salt does she need to add now?

**CEIL Prompt**

Question: A mailman has to give 4 pieces of junk mail to each house in each of the 81 blocks. If there are 12 houses in each blockHow many pieces of junk mail should he give in each block?

A: Let's think step by step There are 12 houses in each block, and the mailman has to give 4 pieces of junk mail to each house. So, in each block, the mailman should give 12 houses x 4 pieces of junk mail = 48 pieces of junk mail. The answer is 48.

Question: After eating a hearty meal they went to see the Buckingham palace. There, Rachel learned that 132 visitors came to the Buckingham palace that day. If 406 people visited the Buckingham palace within the past 327 daysHow many visitors visited the Buckingham palace on the previous day?

A: Let's think step by step: To find out how many visitors visited the Buckingham palace on the previous day, we need to subtract the number of visitors on the current day from the total number of visitors in the past 327 days. So, the number of visitors on the previous day is 406 - 132 = 274. Therefore, 274 visitors visited the Buckingham palace on the previous day. The answer is 274.

Question: There are a total of 16 peaches in a basket. If there are 13 red peaches and some more green peaches in the basket.How many green peaches are in the basket?

A: Let's think step by step If there are a total of 16 peaches in the basket and 13 of them are red, then the remaining peaches must be green. So, the number of green peaches in the basket is 16 - 13 = 3. The answer is 3.

Question: If each bag has 41 cookies and you had 53 bags of cookiesHow many cookies would you have?

A: Let's think step by step If each bag has 41 cookies, and you have 53 bags, then the total number of cookies would be 41 x 53 = 2173 cookies. The answer is 2173.

Question: The Ferris wheel in paradise park has 2 small seats and 23 large seats. Each small seat can hold 14 people and large seat can hold 54 people.How many people can ride the Ferris wheel on small seats?

A: Let's think step by step The number of people that can ride on the small seats is equal to the number of small seats multiplied by the capacity of each small seat. There are 2 small seats and each small seat can hold 14 people, so the total number of people that can ride on the small seats is 2 x 14 = 28. The answer is 28.

Question: Frank was reading through his favorite book. The book had 41 chapters, each with the same number of pages. It has a total of 450 pages. It took Frank 30 days to finish the book.How many pages did he read per day?

A: Let's think step by step. If the book has a total of 450 pages and 41 chapters, then each chapter has 450/41 = 10.975609756097561 pages (rounded to the nearest whole number). Since each chapter has the same number of pages, Frank read approximately 11 pages per chapter. If Frank finished the book in 30 days, then he read 450/30 = 15 pages per day. Therefore, Frank read approximately 15 pages per day. The answer is 15.

Question: Julia played tag with 16 kids on monday. If she played tag with 12 more kids on monday than on tuesdayHow many kids did she play with on tuesday?

A: Let's think step by step: If Julia played with 16 kids on Monday and 12 more kids than on Tuesday, then the number of kids she played with on Tuesday is 16 - 12 = 4. Therefore, Julia played with 4 kids on Tuesday. The answer is 4.

Question: Lewis earns $2 every week during the harvest. If he earns a total of $ 178How many weeks did the harvest last?

A: Let's think step by step: If Lewis earns $2 every week during the harvest, and he earns a total of $178, we can divide the total earnings by the amount he earns per week to find the number of weeks. $178 / $2 = 89 weeks Therefore, the harvest lasted for 89 weeks. The answer is 89.

Table 15: DQ-LoRe Case Study on SVAMP: Part 1

**Problem**

Question: Mary is baking a cake. The recipe calls for 6 cups of flour 8 cups of sugar and 7 cups of salt. She already put in 5 cups of flour.How many more cups of sugar than cups of salt does she need to add now?

**DQ-LoRe (ours)**

Question: Mary is baking a cake. The recipe calls for 5 cups of sugar and 14 cups of flour. She already put in 11 cups of flour.How many more cups of sugar than cups of flour does she need to add now?

A: Let's think step by step. The recipe requires 5 cups of sugar and 14 cups of flour.She already added 11 cups of flour, so she still needs to add 14 - 11 = 3 cups of flour.For the sugar, she needs 5 cups and she hasn't added any yet, so she needs to add all 5 cups.Therefore, she needs to add 5 - 3 = 2 more cups of sugar than cups of flour. The answer is 2.

Question: Mary is baking a cake. The recipe calls for 9 cups of sugar 7 cups of flour and 4 cups of salt. She already put in 2 cups of flour.How many more cups of flour than cups of salt does she need to add now?

A: Let's think step by step. Mary needs to add 7 - 2 = 5 more cups of flour. She needs to add 5 - 4 = 1 more cup of flour than cups of salt. Answer: $\boxed{1}$. The answer is 1.

Question: Mary is baking a cake. The recipe calls for 11 cups of sugar and some more cups of flour. She already put in 3 cups of flour. If she still needs to add 6 more cups of flour.How many cups of flour did the recipe require?

A: Let's think step by step. Mary already put in 3 cups of flour, and she still needs to add 6 more cups of flour. So, the total amount of flour she needs to add is 3 + 6 = «3+6=9»9 cups. Therefore, the recipe requires 9 cups of flour. The answer is 9.

Question: Mary is baking a cake. The recipe calls for 6 cups of sugar and 14 cups of flour. She already put in 7 cups of flour and 60 cups of sugar.How many more cups of flour does she need to add?

A: Let's think step by step. To find out how many more cups of flour Mary needs to add, we need to subtract the amount of flour she already put in from the total amount required. Total cups of flour required = 14 cups Flour already put in = 7 cups To find the remaining amount of flour needed, we do: Remaining flour needed = Total flour required - Flour already put in Remaining flour needed = 14 cups - 7 cups Remaining flour needed = 7 cups Mary needs to add 7 more cups of flour. The answer is 7.

Question: Helen the hippo and her friends are preparing for thanksgiving at Helen's house. Helen baked 519 chocolate chip cookies and 300 raisin cookies yesterday. And she baked 280 raisin cookies and 359 chocolate chip cookies this morning.How many more raisin cookies did Helen bake yesterday compared to today?

A: Let's think step by step. To find out how many more raisin cookies Helen baked yesterday compared to today, we need to subtract the number of raisin cookies she baked today from the number she baked yesterday. Yesterday, Helen baked 519 chocolate chip cookies and 300 raisin cookies, for a total of 519 + 300 = 819 cookies. Today, Helen baked 280 raisin cookies and 359 chocolate chip cookies, for a total of 280 + 359 = 639 cookies. To find the difference in the number of raisin cookies, we subtract the number of raisin cookies baked today from the number baked yesterday: 300 - 280 = 20. Therefore, Helen baked 20 more raisin cookies yesterday compared to today. The answer is 20.

Question: Mary is baking a cake. The recipe calls for 5 cups of flour and 14 cups of sugar. She already put in 10 cups of flour and 2 cups of sugar.How many more cups of sugar does she need to add?

A: Let's think step by step. To find out how many more cups of sugar Mary needs to add, we need to subtract the amount of sugar she already put in from the total amount required. The recipe calls for 14 cups of sugar and Mary already put in 2 cups, so she still needs to add 14 - 2 = 12 more cups of sugar. Therefore, Mary needs to add 12 more cups of sugar. The answer is 12.

The answer is 2.

Table 16: DQ-LoRe Case Study on SVAMP: Part 2

**Problem** Question: Matthew gave equal numbers of crackers and cakes to his 4 friends. If he had 32 crackers and 98 cakes initially. How many crackers did each person eat?

A: Let's think step by step. If Matthew gave equal numbers of crackers and cakes to his 4 friends, then each friend received the same number of crackers and cakes. Let's assume that each friend received x number of crackers and x number of cakes. Since Matthew initially had 32 crackers and he gave an equal number to each friend, the total number of crackers given to his friends is 4x. Similarly, since Matthew initially had 98 cakes and he gave an equal number to each friend, the total number of cakes given to his friends is also 4x. We can set up the equation: 4x = 32 Solving for x, we divide both sides of the equation by 4: x = 32/4 = 8 Therefore, each person ate 8 crackers. The answer is 8.

Question: Rachel had to complete 5 pages of math homework and yet more pages of reading homework. If she had to complete a total of 7 pages of homework. How many pages of reading homework did she have to complete?

A: Let's think step by step. Rachel had to complete 5 pages of math homework and yet more pages of reading homework. If she had to complete a total of 7 pages of homework, then the number of reading homework pages can be found by subtracting the number of math homework pages from the total number of homework pages. So, Rachel had to complete 7 - 5 = 2 pages of reading homework. Therefore, Rachel had to complete 2 pages of reading homework.

