# OpenReview forum: "DQ-LoRe: Dual Queries with Low Rank Approximation Re-ranking for In-Context Learning"
_ICLR.cc/2024/Conference — ICLR 2024 poster_

### Official Review · Reviewer_fk1v · 2023-10-31

**Soundness:** 2 fair
**Presentation:** 3 good
**Contribution:** 2 fair
**Rating:** 5
**Confidence:** 3

**Summary:**

In this paper, the authors introduce a novel framework named DQ-LoRe, aimed at addressing the exemplar selection challenge in Large Language Models (LLMs) for in-context learning. They designed a dual-query mechanism, which first queries the LLM to obtain its generated knowledge and then queries the retriever to obtain the final exemplars. Additionally, they incorporated a low-rank approximation re-ranking technique to ensure that the selected exemplars align closely with the knowledge of the input question. This approach not only focuses on the similarity between the input question and the examples in the training set but also effectively leverages the relationship between the intermediate reasoning steps of the given question and the exemplars. Through a series of experiments, the authors demonstrated the superiority of DQ-LoRe in automatic exemplar selection. Overall, this work offers a fresh perspective on in-context learning and paves the way for future research.

**Strengths:**

1. Originality:
The paper introduces a novel framework named DQ-LoRe (Dual Queries with Low Rank Approximation Re-ranking) that addresses the challenge of selecting exemplars for in-context learning.
The approach of using Dual Queries is innovative. It first queries the Large Language Model (LLM) to obtain LLM-generated knowledge such as Chain-of-Thought (CoT) and then queries the retriever to obtain the final exemplars using both the question and the knowledge.
The concept of using Low Rank Approximation for re-ranking, especially in the context of in-context learning, adds to the originality of the work.

2. Quality:
The paper showcases extensive experiments to validate the effectiveness of DQ-LoRe.
The results indicate that DQ-LoRe significantly outperforms prior state-of-the-art methods in the automatic selection of exemplars for GPT-4.
The paper provides a comprehensive analysis, revealing that DQ-LoRe consistently surpasses retrieval-based approaches in terms of both performance and adaptability.

3. Clarity:
The paper is well-structured and presents its methodology and findings in a clear and concise manner.
The use of terms like Dual Queries, Low-rank approximation Re-ranking, and Chain-of-Thought are well-defined and contribute to the clarity of the paper's content.

4. Significance:
The paper addresses a central challenge in the domain of in-context learning, which is the effective selection of exemplars. The improvement in performance from 92.5% to 94.2% on GSM8K highlights the significance of the proposed approach. By pushing the boundaries of in-context learning, DQ-LoRe opens up new avenues for addressing complex reasoning challenges, making it a significant contribution to the research community.

In summary, the paper presents a significant and original contribution to the field of in-context learning by introducing the DQ-LoRe framework. The quality of the research is evident from the comprehensive experiments and analysis provided, and the content is presented with clarity.

**Weaknesses:**

1. Limited Originality:
While the paper introduces the DQ-LoRe framework, the methods employed (such as BM25, PCA dimensionality reduction, etc.) are based on prior research. Although the authors have adeptly integrated and applied these methods within their framework, from an innovation standpoint, these techniques are not novel in themselves.

2. Methodological Concerns:
The first round of sorting in the paper uses BM25 to match the question with the question in the exemplar, aiming to find the exemplar most relevant to the question. The second round aims to find the Chain-of-Thought (CoT) most relevant to the question. This raises a question: Why not directly compute the similarity between the CoT in the exemplar and the CoT of the question itself? Instead, the approach seems to add unnecessary complexity by opting to compute scores for the generated CoT.

3. Experiments Could Be More Robust:
While the authors perform commendably in comparative experiments, the ablation studies appear to be lacking. For instance, it would be beneficial to compare the effects of two rounds of sorting versus a single round, and the effects of PCA dimensionality reduction versus no reduction. Such experiments might offer readers more insights into the efficacy of the methods.

4. Detailing Issues:
Figure 1 in the paper has errors. Specifically, in the "M Q&A Low rank embedding" section, "Embedding 1" appears twice, which might lead to confusion. Authors should ensure that all charts and graphics are accurate and clear to avoid any potential misunderstandings.

**Questions:**

1. Originality Concerns:
Could the authors elaborate on the novelty of the DQ-LoRe framework, especially considering that methods like BM25、PCA、text-davinci-003 dimensionality reduction have been previously employed in other works?

2. Methodological Clarifications:
What was the rationale behind opting for two rounds of sorting (first for matching the question with the exemplar and second for finding the relevant CoT) instead of directly computing the similarity between the CoT in the exemplar and the CoT of the question?

3. Experimental Details:
In the ablation studies, could the authors provide results comparing the effects of two rounds of sorting versus a single round? Additionally, it would be beneficial to see the effects of PCA dimensionality reduction versus no reduction. How do these individual components impact the overall performance?

---

> ### Author Response · Authors · 2023-11-16
> **Response to Reviewer fk1v (Part 1/3)**
>
> We used GPT-3.5-turbo-16K as the engine for experiments in the following study.
>
> **Question1:Limited Originality: While the paper introduces the DQ-LoRe framework, the methods employed (such as BM25, PCA dimensionality reduction, etc.) are based on prior research. Although the authors have adeptly integrated and applied these methods within their framework, from an innovation standpoint, these techniques are not novel in themselves.**
>
> **Response:**
>
> Thank you for your constructive feedback. We would like to address some misunderstandings highlighted in the paper and have incorporated additional experiments based on your recommendations to further substantiate the efficacy of our approach.
>
> To begin with, we need to clarify that BM25 is not employed in our principal methodology, DQ-LoRe, as delineated in our publication. Its use is confined to the training phase for generating positive and negative sample pairs.
>
> As illustrated in Figure 1, the second query retrieval using the smaller model employs the encoder’s output, which is trained on CoT+Question. In this stage, we compute the embedding of the current CoT (derived from the initial query) combined with the Question (from the current test set). We then identify the $M$ samples from the training set’s CoT+Question embeddings that most closely resemble the current test sample's question and Chain-of-Thought. Here, $M$ exceeds the final number of exemplars, $N$. This is because we have observed significant spurious correlation issues in methods relying on the initial retrieval, such as EPR. Therefore, within the initial set of $M$ samples, we seek to eliminate this portion through denoising techniques. We achieve this by employing PCA as a denoising method to project these $M$ samples into a lower-dimensional space and then re-ranking them to obtain the final set of $N$ exemplars, thus mitigating the interference from spurious correlation samples. **Notably, BM25 is not used in this initial sorting phase during inference.**
>
> While our framework builds upon existing technologies, it stems from a conceptually robust idea: enhancing the precision of the $M$ nearest exemplars (sourced from the dual-query process with embeddings encapsulating both Question and Chain-of-Thought data) by transposing them into a novel vector space and re-ranking them through denoising to pinpoint the Top $N$ most pertinent exemplars. This refinement is vital for improved in-context learning.
>
> The integration of other technologies is pivotal to this overarching concept. For example, we utilized PCA for enhanced sample differentiation. Although Gaussian kernel methods could theoretically be applied for dimensionality expansion in high-dimensional space to separate linearly inseparable samples, they did not produce satisfactory outcomes in our experiments.
>
> Moreover, to guarantee the comprehensiveness and sufficiency of the embedding data before implementing PCA, we initially query the larger model with a combination of the initial exemplar and the current question to gather a priori information. Subsequently, we merge the CoT and question to query the smaller model. Therefore, the preliminary querying phase with the larger model aims to procure the CoT, ensuring that the embeddings derived from the second query to the smaller model are rich in information, encompassing both CoT and Question elements. This rationale underpins our Dual Query methodology.

---

> ### Author Response · Authors · 2023-11-16
> **Response to Reviewer fk1v (Part 2/3)**
>
> **Question2:Methodological Concerns: The first round of sorting in the paper uses BM25 to match the question with the question in the exemplar, aiming to find the exemplar most relevant to the question. The second round aims to find the Chain-of-Thought (CoT) most relevant to the question. This raises a question: Why not directly compute the similarity between the CoT in the exemplar and the CoT of the question itself? Instead, the approach seems to add unnecessary complexity by opting to compute scores for the generated CoT.**
>
>
> **Response:**
>
> Thank you for your constructive feedback.
>
> Firstly, we wish to clarify that BM25 was not utilized for the first-round ranking to obtain sorted samples. Its application was limited to the encoder training phase, where it played a crucial role in constructing positive and negative training samples.
>
> In a similar vein, the computation of scores for the generated Chain-of-Thought (CoT) was instrumental in encoder training, aiding in the creation of positive and negative samples. This scoring mechanism was not employed during the inference stage for re-ranking purposes.
>
> During training, constrained by resources, we leveraged BM25 to select a narrowed subset of samples for scoring. This scoring process was integral to forming the definitive positive and negative samples. More specifically, in the encoder's training, BM25 facilitated a preliminary screening to identify the top n sentences from examples semantically akin to the input question. These chosen sentences, along with the question, were then presented to a smaller model. The confidence level of the smaller model’s response served as an indicator of the sentence’s usefulness in addressing the question.
>
> Post contrastive learning training, our encoder adeptly clusters the most beneficial examples for a question in a high-dimensional space, closely aligning them with the question.
>
> **It's important to note that these techniques were exclusive to the encoder training phase and were not a part of the two-stage ranking process during inference.**
>
> In the initial ranking phase, we utilized encoded representations of the current test sample's question combined with its CoT (from the first query). These were sorted based on their similarity to the encoded representations of question + CoT from all training samples, integrating both CoT and question information.
>
> Furthermore, we embarked on experiments that concentrated exclusively on the similarity between the CoTs of the current test sample and those of the training samples. We also expanded our scope to include additional ablation experiments, which aimed to dissect and understand the distinct contributions of question information and Chain-of-Thought (CoT) information. This was especially pertinent in the context of employing LoRe.
>
> Table 1: Comparing the Use of Trained Encoder and Solely CoT Similarity Retrieval.
>
> |        | w/o trainning |  with trainning  |
> |  ----  | :----:  | :----:  |
> | question + CoT  | 77.7 | 78.9 |
> | question + CoT with LoRe  | 78.8 | 80.7 |
> | only CoT  | 76.1 | 79.4 |
> | only CoT with LoRe  | 78.1 | 80.4 |
>
>
> * The "w/o training" denotes the direct use of an untrained BERT for encoding sample representations.
> * The "with training" denotes the use of a BERT model trained on a training set for encoding sample representations.
> * The "question + CoT" denotes the simultaneous use of both question information and the information from CoT.
> * The 'only CoT'  denotes to exclusively using 'CoT' information to encode sample representations. **This corresponds to directly computing the similarity between the CoT in the exemplar and the CoT of the question itself.**
> * The 'LoRe' denotes the low-rank approximation re-ranking algorithm that we have proposed.
>
> In this experiment, we investigate the effects of training the encoder, not training the encoder, and employing single-round and two-round ranking on the final results. By comparing the 'without training' and 'with training' settings, it has been observed that training the encoder on specific datasets significantly enhances the retrieval task performance in certain datasets. This is particularly evident in the 'only CoT' setting, where there is a notable performance difference of nearly *3.3%*. Furthermore, our approach exhibits robustness, as it achieves consistent improvements even without employing an encoder trained on a specific dataset. Especially in the 'only CoT with LoRe' setting under 'w/o training', we achieve an absolute performance improvement of *2%* without specifically training the encoder. Finally, we discovered that using only CoT similarity yields inferior results compared to utilizing similarity based on both Question and CoT.

---

> ### Author Response · Authors · 2023-11-16
> **Response to Reviewer fk1v (Part 3/3)**
>
> **Question3:Experiments Could Be More Robust: While the authors perform commendably in comparative experiments, the ablation studies appear to be lacking. For instance, it would be beneficial to compare the effects of two rounds of sorting versus a single round, and the effects of PCA dimensionality reduction versus no reduction. Such experiments might offer readers more insights into the efficacy of the methods.**
>
> **Response:**
>
> Thank you for your valuable suggestions. As per your recommendations, we have conducted additional comparative experiments, which we outline below.
> To begin, we compared the impact of various re-ranking methods. 'w/o re-ranking' serves as our baseline, where no re-ranking was performed. 'Longest' represents a re-ranking approach that extends the Complex-CoT method. In this method, exemplars are selected for the final inference step based on the longest 8 CoT exemplars from the initial retrieval of 16 or 32 exemplars (the selection of 16 or 32 exemplars is a tunable hyperparameter).
> In addition to these, we conducted experiments employing BERT-Whitening[1], MGS (Modified Gram-Schmidt Process)[2], and Gaussian Kernel[3] for re-ranking. The detailed experimental results are presented in the table below:
>
> Table 2: Comparison of Different Inference LLMs and Ablation of Re-Ranking Algorithms
> |    Re-ranking method    | Text-Davinci-003 | GPT3.5-Turbo-16k
> |  ----  | :----:  | :----:  |
> | w/o re-ranking  |67.9| 78.9 |
> | Longest  |68.1| 78.9 |
> | BERT-Whitening  |67.9| 78.9 |
> | MGS  |68.2| 79.7 |
> |Gaussian Kernel  |66.7| 79.0 |
> | PCA  |69.1| 80.7 |
>
>
> As illustrated in the table above, we establish a baseline by omitting the re-ranking step and employing a single-round ranking model. Subsequently, we undertake a comprehensive analysis of the impact of employing various algorithms for the second-round ranking. It is noteworthy that only two methods, namely MGS and PCA, exhibit a discernible impact on the re-ranking process. This suggests that the effectiveness of the re-ranking process is attributed not only to dimensionality reduction of vectors but also to the orthogonalization of vectors, both playing significant roles.
> In contrast, kernel tricks such as the Gaussian Kernel do not appear to contribute significantly to the sample distinction in this context. Notably, in the case of text-davinci-003, we observed a performance decline of 1.2%.
>
> Table 3: Experiments with Different Encoders and Ablation of LoRe Across Baselines
> |    Method    | BERT-base-uncased | RoBERTa-base |
> |  ----  | :----:  | :----:  |
> | EPR |77.3| 78.0 |
> | EPR + LoRe  | 77.0| 77.8 |
> | CEIL  |79.4| 77.7 |
> | CEIL + LoRe  |78.7 | 77.4 |
> | Dq-LoRe w/o LoRe | 78.9 | 76.4 |
> | Dq-LoRe  |80.7| 78.8 |
>
> In the second experiment presented in the table, we investigate the effects of utilizing different base models on the experimental outcomes. Our findings indicate that, in terms of overall performance, the bert-base-uncased model outperforms the RoBERTa-base model, except for an enhancement observed in EPR when employing the RoBERTa-base.
> Furthermore, we assess the influence of LoRe on various models and note that the adoption of LoRe leads to a decline in performance for both EPR and CEIL. This decline can be primarily attributed to the absence of the Dual Query process, which hampers the encoding of additional Chain-of-Thought (CoT) information.
>
> **References:**
>
> >[1] Su, J., Cao, J., Liu, W., & Ou, Y. (2021). Whitening sentence representations for better semantics and faster retrieval. arXiv preprint arXiv:2103.15316.
>
> >[2] Longley, J. W. (1981). Modified gram-schmidt process vs. classical gram-schmidt: Modified gram-schmidt process vs. classical gram-schmidt. Communications in Statistics-Simulation and Computation, 10(5), 517-527.
>
> >[3] Scholkopf, B., Sung, K. K., Burges, C. J., Girosi, F., Niyogi, P., Poggio, T., & Vapnik, V. (1997). Comparing support vector machines with Gaussian kernels to radial basis function classifiers. IEEE transactions on Signal Processing, 45(11), 2758-2765.
>
>
> **Question4:Detailing Issues: Figure 1 in the paper has errors. Specifically, in the "M Q&A Low rank embedding" section, "Embedding 1" appears twice, which might lead to confusion. Authors should ensure that all charts and graphics are accurate and clear to avoid any potential misunderstandings.**
>
> **Response:**
>
> Thank you for pointing out these mistakes, we will revise them accordingly in the rebuttal version, which will be submitted later this week.

---

> ### Author Response · Authors · 2023-11-21
> **Waiting for further discussion.**
>
> Dear Reviewers,
>
> Thank you very much for your invaluable feedback on our paper. We have meticulously reviewed each of your points and endeavored to address them thoroughly. We would greatly appreciate it if you could review our responses and let us know if you have any additional comments regarding the paper or the rebuttal. We are eager to embrace all your critiques and integrate them into our work.
>
> Thank you!

---

### Official Review · Reviewer_FmMb · 2023-11-01

**Soundness:** 3 good
**Presentation:** 3 good
**Contribution:** 3 good
**Rating:** 6
**Confidence:** 3

**Summary:**

In this paper, authors propose a new approach to extract exemplars in LLM for effective inference. They propose a 2 stage process where first stage focuses on chain of thought reasoning for a given query, the chain of thought output is then used with dimentionality reduced model to extract similar queries which is then used to perform effective inference. They show improvement over previous approaches is choosing exemplars.

**Strengths:**

Well written paper. Easy to understand. Simple yet effective approach to improve prompt generation and model performance of LLMs.

**Weaknesses:**

Novelty not high in my opinion. Low-rank approximation is a neat extension, but it is being used as regularizer to some extent. No mention of any other form of regularization is mentioned.

**Questions:**

To support statements like low-rank helps mitigate finding spurious correlation, it would be good to include a version of their approach in the experiment where low-rank approximation was not performed.
Minor but makes it a bit hard to read: Too many abbreviation and some like ERP are not expanded (assuming they consider it as background knowledge). Not much mention of what the baselines are. Will be good to include that a bit.

---

> ### Author Response · Authors · 2023-11-20
> **Response to Reviewer FmMb (Part 1/2)**
>
> **Question1:Novelty not high in my opinion. Low-rank approximation is a neat extension, but it is being used as regularizer to some extent. No mention of any other form of regularization is mentioned.**
>
> **Response:**
> Thank you for your feedback on our work and your appreciation of low-rank approximation. We agree that low-rank approximation itself is not a new idea. However, the approach of first identifying similar exemplars, then mapping them to another vector space (using low-rank approximation here), and finally re-ranking to obtain the ultimate exemplars is innovative. To the best of my knowledge, no prior works have utilized such a strategy to retrieve exemplars for better in-context learning.
>
> Specifically, our approach involves retrieving $M$ most similar exemplars from the entire training set, mapping these $M$ samples to another embedding space, and then calculating the similarity with the question+CoT of the current test set. The re-ranking process yields the final $N$ exemplars ($M$>$N$). This idea is not only clever but also innovative.
>
> Regarding the issue of 'No mention of any other form of regularization is mentioned,' we present relevant experiments below:
>
> Table 1: Supplementary experiments and ablation studies are conducted with different regularizations in the re-ranking algorithm.
> |    re-ranking method    | GSM8K |  SVAMP  |  AQUA  |  StrategyQA  |  QASC  |
> |  ----  | :----:  | :----:  | :----:  | :----:  | :----:  |
> | w/o re-ranking   | 78.9 | 89.0 | 57.0 | 74.1 | 82.1 |
> |  Gaussian kernel  | 79.0 | 87.6 | 56.2 | 75.4 | 82.7 |
> | MGS      | 79.7 | 87.3 | 57.8 | 73.2 | 81.0. |
> | ICA      | 79.5 | 85.7 | 57.0 | 74.2 | 82.2 |
> | PCA      | 80.7 | 90.0 | 59.8 | 74.6 | 81.2 |
> | bert-base-uncased   | 80.7 | 90.0 | 59.8 | 75.4 | 82.7 |
> | roberta-base    | 78.8 | 89.0 | 57.8 | 74.2 | 81.2 |
>
> * The 'w/o re-ranking' serves as our baseline, where no re-ranking was performed. This means that the algorithm performs sorting only once after querying LLMs.
> * The  'Gaussian kernel' [1] denotes the calculation of similarity after mapping the embeddings of exemplars to a high-dimensional space during our re-ranking process.
> * The 'MGS' (Modified Gram-Schmidt Process) [2] denotes the mapping of exemplars' embeddings to an orthogonal basis during our re-ranking process.
> * The  'ICA' (Independent Component Analysis) [3] denotes a theoretically guaranteed disentangled representation, emphasizing the ability to recover latent variables (i.e., identifiability) in the data. Here, we map the feature dimensions of the embedding to 16 dimensions.
> * "bert-base-uncased" [4] and "roberta-base" [5] denote the use of different encoders for the similarity calculation in the first-stage ranking.
> From the above results, we can observe that the choice of different regularization techniques has a significant impact on the re-ranking algorithm. Specifically, we observed performance improvement in Math Word Problems when using PCA and MGS. On the other hand, for Q&A tasks, the Gaussian kernel method showed a substantial performance boost. In Q&A tasks, there is a notable distinction compared to solving math problems. The background knowledge needed for Math Word Problems is embedded within the question itself, whereas factual information for Q&A tasks requires querying from Large Language Models (LLMs). Additionally, Q&A tasks are sensitive to textual details. Therefore, filtering information in embeddings is not advisable for Q&A tasks, as dimensionality reduction could eliminate crucial entity information. Hence, we have employed a kernel method to transform the embeddings, ensuring that information is retained within the embeddings while still becoming separable.
> From the above results, we can draw a conclusion that the regularization method used before re-ranking has a significant impact on the final performance, depending on the specific task.
>
> **Reference:**
> >[1] Scholkopf, B., Sung, K. K., Burges, C. J., Girosi, F., Niyogi, P., Poggio, T., & Vapnik, V. (1997). Comparing support vector machines with Gaussian kernels to radial basis function classifiers. IEEE transactions on Signal Processing, 45(11), 2758-2765.
>
> >[2] Longley, J. W. (1981). Modified gram-schmidt process vs. classical gram-schmidt: Modified gram-schmidt process vs. classical gram-schmidt. Communications in Statistics-Simulation and Computation, 10(5), 517-527.
>
> >[3] Hyvärinen, A., & Oja, E. (2000). Independent component analysis: algorithms and applications. Neural networks, 13(4-5), 411-430.
>
> >[4] Devlin, J., Chang, M. W., Lee, K., & Toutanova, K. (2018). Bert: Pre-training of deep bidirectional transformers for language understanding. arXiv preprint arXiv:1810.04805.
>
> >[5] Liu, Y., Ott, M., Goyal, N., Du, J., Joshi, M., Chen, D., ... & Stoyanov, V. (2019). Roberta: A robustly optimized bert pretraining approach. arXiv preprint arXiv:1907.11692.

---

> ### Author Response · Authors · 2023-11-20
> **Response to Reviewer FmMb (Part 2/2)**
>
> **Question2:To support statements like low-rank helps mitigate finding spurious correlation, it would be good to include a version of their approach in the experiment where low-rank approximation was not performed. Minor but makes it a bit hard to read: Too many abbreviation and some like ERP are not expanded (assuming they consider it as background knowledge). Not much mention of what the baselines are. Will be good to include that a bit.**
>
> **Response:**
>
> Thank you for your valuable feedback. In the new experimental table, we have included other re-ranking algorithms that do not perform low-rank approximation. Additionally, we have conducted new ablation experiments to demonstrate the effectiveness of our method. The specific results are presented in the aforementioned Table 1 and the following Tables 2 and 3:
>
> Table 2: Comparison of Different Inference LLMs and Ablation of Re-Ranking Algorithms in GSM8K.
> |    Re-ranking method    | Text-Davinci-003 | GPT3.5-Turbo-16k |
> |  ---- | :----:  | :----:  |
> | w/o re-ranking  |67.9| 78.9 |
> | Longest  |68.1| 78.9 |
> | BERT-Whitening  |67.9| 78.9 |
> | MGS  |68.2| 79.7 |
> |Gaussian Kernel  |66.7| 79.0 |
> | PCA  |69.1| 80.7 |
>
> Table 3: Experiments with Different Encoders and Ablation of LoRe Across Baselines in GSM8K.
> | Method| BERT-base-uncased | RoBERTa-base |
> |  ---- | :----:  | :----:  |
> | EPR |77.3| 78.0 |
> | EPR + LoRe  | 77.0| 77.8 |
> | CEIL  |79.4| 77.7 |
> | CEIL + LoRe  |78.7 | 77.4 |
> | Dq-LoRe w/o LoRe | 78.9 | 76.4 |
> | Dq-LoRe  |80.7| 78.8 |
>
> Table 4: Comparing whether to adopt a trained encoder in GSM8K.
> |        | w/o trainning |  with trainning  |
> |  ----  | :----:  | :----:  |
> | DQ-LoRe w/o LoRe  | 77.7 | 78.9 |
> | DQ-LoRe | 78.8 | 80.7 |
>
> Our specific experimental results are shown in the table above. Performing low-rank approximation directly on LoRe and CEIL does not work. We also discovered in experiments that it is necessary to add the Dual-Query module to the retriever for our method to be effective. The detailed experimental results are presented in Table 4. We found that our method works even when using an untrained encoder. Additionally, not performing low-rank re-ranking significantly decreases the model's performance, demonstrating the effectiveness of our approach.
> In response to your question, "Not much mention of what the baselines are. Will be good to include that a bit," we commit to describing our baselines and expanding the abbreviations in rebuttal versions.

---

> ### Author Response · Authors · 2023-11-21
> **Waiting for further discussion.**
>
> Dear Reviewers,
>
> Thank you very much for your invaluable feedback on our paper. We have meticulously reviewed each of your points and endeavored to address them thoroughly. We would greatly appreciate it if you could review our responses and let us know if you have any additional comments regarding the paper or the rebuttal. We are eager to embrace all your critiques and integrate them into our work.
>
> Thank you!

---

> > ### Comment · Reviewer_FmMb · 2023-11-23
> > **Thank you for your response**
> >
> > Thank you for the detailed response and additional experiments. They are helpful in understanding the approach better. Thanks

---

### Official Review · Reviewer_vwFb · 2023-11-03

**Soundness:** 3 good
**Presentation:** 3 good
**Contribution:** 3 good
**Rating:** 6
**Confidence:** 4

**Summary:**

The paper proposes DQ-LoRe, a method of completing reasoning tasks by retrieving higher quality in-context exemplars. In the first query, the question and initial exemplars are used to induce a CoT from the LLM, which is then used in the second query to retrieve the final exemplars. Those exemplars are then used as part of the prompt for the final inference to complete the tasks. The retriever is trained on BM25 and LLM predictions. PCA is used to reduce the embedding dimensions so that the retrieval can be based on non-spurious features.

Using GPT-4 as its engine, the method reaches state-of-the-art performance on GSM8K and shows strong out-of-domain performance. Ablation, visualization, and case studies are included to help better understand the method.

**Strengths:**

- The method achieves SOTA performance as well as great robustness on an important area of current NLP/LLM research: reasoning.
- The core method, DQ-LoRe is original to the best of my knowledge. It is also relatively easy to understand.
- The method is implemented at the prompt level, so it should be easy to apply to a wide range of models and use cases.
- The presentation of the study is good. Figure 1 is clear and helpful. The paragraphs are generally well written. See typos in the Questions section.

**Weaknesses:**

- The return on Investment (ROI) of the method is not super high. The method outperforms SOTA accuracy by 1.7 at the cost of：
 1. Complex implementation. The method involves an additional query of the LLM, a retrieving step, a dimensionality reduction step, and a re-ranking step. One might find it hard to justify the complexity with the 1.7 point improvement.
2. High latency, as a result of the additional steps, which makes it not ideal for real-life use cases.
- Reproducibility: For complex systems like this, it is important for the authors to release their source code subsequent studies can make use of. I do not find a promise to release the source code. Will you release your code?
- Would be great to include more LLMs to demonstrate the generalizability of the method.

**Questions:**

- Typo? on page 9: "a dual-query framework that enhances in-context learning for multi-step reasoning tasks by considering the Contexts of Interest (CoTs) in input questions and exemplars"
- Typo on page 2: "Following the acquisition of re-ranked exemplars from the smaller retrieval model, DQ-LoRe subsequently provides the exemplars to the LLMs for inference.,"
- Would be nice to include an analysis of latency.

---

> ### Author Response · Authors · 2023-11-22
> **Response to Reviewer vwFb (Part 1/4)**
>
> **Question1:Typo? on page 9: "a dual-query framework that enhances in-context learning for multi-step reasoning tasks by considering the  (CoTs) in input questions and exemplars"**
>
> **Response:**
> Thank you for pointing out these mistakes, we have revised accordingly in the rebuttal version, which will be submitted later this week.
>
> **Question2:Typo on page 2: "Following the acquisition of re-ranked exemplars from the smaller retrieval model, DQ-LoRe subsequently provides the exemplars to the LLMs for inference."**
>
> **Response:**
> Thank you for pointing out these mistakes, we have revised accordingly in the rebuttal version, which will be submitted later this week.
>
> **Question3:For complex systems like this, it is important for the authors to release their source code subsequent studies can make use of. I do not find a promise to release the source code. Will you release your code?**
>
> **Response:**
> Of course, we will release our source code. We are currently in the process of organizing our final code. If our paper is accepted, you will be able to find the details in our camera-ready version.

---

> ### Author Response · Authors · 2023-11-22
> **Response to Reviewer vwFb (Part 2/4)**
>
> **Question4:Would be nice to include an analysis of latency.**
>
> **Response:**
> Thank you for your valuable feedback. We appreciate the constructive insights you provided. In response to the issues you raised, we have taken steps to clarify certain points in the paper.
>
> Our methodology encompasses two distinct latency stages: training and inference. In the ensuing discussion, we aim to dispel the notion that our approach is 'not ideal for real-life use cases,' a viewpoint we consider inaccurate.
>
> Firstly, let's consider the latency during the training of the small model.    In practice, a single training iteration often proves sufficient for a given dataset type.    We deem this to be an acceptable cost in both academic and industrial contexts.    Additionally, as illustrated by the experiments outlined in Table 1 of our paper, our small model trained on the GSM8K dataset exhibits outstanding performance when applied to the SVAMP dataset.    This underscores the robustness of our method, offering a notable advantage in real-world applications.
>
> Moving on to the second aspect of latency during the inference stage, we conducted thorough tests to compare the time efficiency of our approach against baseline methods.    Subsequently, we conducted a detailed analysis of the time consumption for each specific module within our method.    For a more comprehensive understanding, please refer to the detailed experimental results provided in the subsequent table.
>
> Table 1: Average Processing Time per Question for DQ-LoRe and Baseline
> |   Method    | EPR | CEIL| DQ-LoRe|
> |  ----  | :----:  | :----:  | :----:  |
> | Time  |0.756s| 0.758s | 1.584s |
>
>
>
> Table 2: Detailed Time Consumption of Each Stage in DQ-LoRe
> |  DQ-LoRe  | First Query | Second Query |  Re-Ranking |Inference
> |  ----  | :----:  | :----:  | :----:  | :----:  |
> | Time  |0.728s| 0.028s | 0.101s | 0.727s
>
> From Table 1, it is evident that the time expended by our method exhibits a linear relationship in comparison to the baseline. Further insights from Table 2 reveal that the predominant additional consumption, in contrast to the baseline, occurs during the initial stage of requesting the large model to acquire chain-of-thoughts (CoTs). (In practice, each API key for the 3.5-16k-turbo model can process 180,000 tokens per minute, and with an average request requiring around 2000 tokens, theoretically, by employing numerous keys and parallelizing threads, the time consumption in this stage can be significantly reduced.)

---

> ### Author Response · Authors · 2023-11-22
> **Response to Reviewer vwFb (Part 3/4)**
>
> **Question5:The return on Investment (ROI) of the method is not super high. The method surpasses the state-of-the-art accuracy by 1.7, albeit at a cost.**
>
> **Response:**
>
> Exactly what we want is to propose a flexible "component" designed to seamlessly integrate with most methods. The experiments in our paper are orthogonal to works such as EPR and CEIL. We conducted ablation studies based on their methodologies to demonstrate a substantial improvement relative to the baselines. In practical applications using GPT-4 on the GSM8K dataset, we achieved a 2.9-point improvement, elevating performance from 91.3 to 94.2. In Table 3 below, we migrate our method to CEIL and conduct ablation experiments on each step to provide insights and address concerns related to the “The return on Investment (ROI) of the method is not super high.”.
>
> Table 3: Ablation Experiments on Individual Components in EPR and CEIL
> |   Method    | GPT3.5-Turbo-16k | GPT-4|
> |  ----  | :----:  | :----:  |
> | EPR  |77.3| 91.3 |
> | EPR + DQ |78.3| 93 |
> | EPR + DQ-LoRe  |80.7| 94.2 |
> | CEIL  |79.4| 92.5 |
> | CEIL + LoRe  |78.7| 92 |
> | CEIL + DQ  |79.3| 92.3 |
> | CEIL + DQ-LoRe  |79.9| 94.1 |
>    * The "Method + DQ" denotes that we use the Dual Queries and simultaneous use of both question information and the information from CoT.
>    * The "Method + LoRe" denotes the utilization of Low-Rank Approximation Re-ranking, exclusively relying on question information without the information from CoT.
>    * The "Method + DQ-LoRe" denotes that we use the DQ-LoRe.

---

> ### Author Response · Authors · 2023-11-22
> **Response to Reviewer vwFb (Part 4/4)**
>
> **Question6:Would be great to include more LLMs to demonstrate the generalizability of the method.**
>
>
> **Response:**
>
> Thank you for your valuable suggestions. As per your recommendations, we have conducted additional comparative experiments using Llama2-7b-hf, Llama2-7b-chat-hf and Vicuna, which can be found in Table 4.
>
> Table 4: Experiments on more LLMs
> |   Method    | Llama2-7b-hf | Llama2-7b-chat-hf| Vicuna-7b
> |  ----  | :----:  | :----:  | :----:  |
> | Cot  			|12.6| 22.9 | 19.4 |
> | Complex Cot 	|17.7| 29.4 | 23.8 |
> | Auto-Cot  	|15.0| 27.0 | 23.9 |
> | EPR  			|15.1| 23.0 | 22.0 |
> | CEIL  		|15.2| 26.7 | 22.4 |
> | DQ-LoRe  		|16.0| 28.9 | 23.8 |
>
> We observe that compared to the retrieval baselines (EPR and CEIL), our method demonstrates significant improvements even on models with smaller open-source parameter sizes. Moreover, the effectiveness of our approach increases with the improvement in the model's instruction-following capabilities. However, we also acknowledge that, when compared to Complex cot and Auto-cot, our method does not show as significant improvements on the LLaMa family 7b-scale models.

---

### Official Review · Reviewer_QTwu · 2023-11-04

**Soundness:** 3 good
**Presentation:** 3 good
**Contribution:** 3 good
**Rating:** 6
**Confidence:** 4

**Summary:**

In this paper, the authors propose to leverage dual queries and low-rank approximation re-ranking to find the exemplars for in-context learning. LLM-generated knowledge can first be derived by dual queries so that the retriever can provide the final exemplars with both the question and the acquired knowledge. Experiments are conducted on several benchmark datasets when some of the datasets involve chain-of-thought (CoT) annotations. The experimental results show that the proposed method can outperform several conventional in-context learning methods with GPT-4 in the in-domain setup. With domain shifts, the proposed framework also surpasses baseline methods with two different LLM engines. Besides, the authors also conduct some ablation and analysis studies to show the effectiveness of the key component LoRE.

**Strengths:**

* S1: The LoRe component can significantly improve the performance of models that consider CoT as shown in the ablation study.
* S2: The improvements are consistent within most of the cases for both in-domain and domain-shifted scenarios.
* S3: Ablation and qualitative studies demonstrate the rationales behind the improvements.

**Weaknesses:**

* W1: The idea of "dual queries" is not novel when many studies [a] have already utilized LLMs themselves to have better queries for retrieval augmentation.
* W2: Datasets are limited. All of the datasets are about arithmetic questions.
* W3: Some mentioned related studies like Auto-CoT (Zhang et al., 2022) are not compared in the experiments, especially while their studies are more general and conducted on more datasets.


[a] Xu, S., Pang, L., Shen, H., Cheng, X., & Chua, T. S. (2023). Search-in-the-Chain: Towards the Accurate, Credible and Traceable Content Generation for Complex Knowledge-intensive Tasks. arXiv preprint arXiv:2304.14732.

**Questions:**

* Q1: I wonder if the authors conduct significance tests on the improvements of the proposed method over baseline methods.
* Q2: Following W2 and W3, it would be great if the authors could involve more datasets and baseline methods in the experiments.Q4:

**Details Of Ethics Concerns:**

N/A.

---

> ### Author Response · Authors · 2023-11-19
> **Response to Reviewer QTwu (Part 1/3)**
>
> **Question1:The idea of "dual queries" is not novel when many studies [a] have already utilized LLMs themselves to have better queries for retrieval augmentation.**
>
> **Response:**
> Thank you for the insightful question.
>
> Firstly, we need to state the difference between our motivations. In fact, our motivation behind interacting LLM with IR differs significantly from what is presented in the paper [1]. Our initial query to the LLM aims to gather more information to enrich the LoRe model for the second query. Our ablation studies clearly demonstrate that without the CoT information supplemented by the first LLM query, our second query using LoRe is ineffective. The specific experimental results are shown in:
>
> Table 1: Experiments with Different Encoders and Ablation of LoRe Across Baselines
> |    Method    | BERT-base-uncased | RoBERTa-base |
> |  ----  | :----:  | :----:  |
> | EPR |77.3| 78.0 |
> | EPR + LoRe  | 77.0| 77.8 |
> | CEIL  |79.4| 77.7 |
> | CEIL + LoRe  |78.7 | 77.4 |
> | Dq-LoRe w/o LoRe | 78.9 | 76.4 |
> | Dq-LoRe  |80.7| 78.8 |
>
> From the experiments above, we can observe that, in fact, the LoRe module does not work on EPR and CEIL without the interaction process between LLM and IR. There is even a performance decrease of 0.2% and 0.3%, respectively, due to the lack of necessary CoT information for re-ranking, necessitating an additional query to LLMs. **This experimental result provides an important insight that Dual Query is a prerequisite for the effective functioning of LoRe.**
>
> Secondly, the content we retrieve in IR differs; we use IR to retrieve exemplars necessary for in-context learning, rather than knowledge. Our knowledge is supplied and augmented by LLMs within the IR retrieval process. In contrast, literature [1] employs the retrieval of factual knowledge from IR to enhance Large Language Models (LLMs), consequently diminishing the occurrence of hallucinations.
>
> Lastly and most importantly, the interaction between Information Retrieval (IR) and Large Language Models (LLMs) is not our core contribution. We merely consider the Dual-Query as a baseline for the LoRe method, which is fundamental for making LoRe work. Article [1] uses IR to provide validation for LLMs and infuses knowledge from IR into LLMs, where IR acts as an auxiliary tool external to LLMs. In our work, we use the Dual-query approach to treat LLMs as an auxiliary knowledge base for IR, allowing IR to acquire necessary information from LLMs to inject into IR's embeddings for dimensionality reduction and re-ranking. It is evident that, although both methods involve interactions between LLMs and IR, their specific implementation goals and details are entirely different.
>
> **Reference**
> >[1] Xu, S., Pang, L., Shen, H., Cheng, X., & Chua, T. S. (2023). Search-in-the-Chain: Towards the Accurate, Credible and Traceable Content Generation for Complex Knowledge-intensive Tasks. arXiv preprint arXiv:2304.14732.

---

> ### Author Response · Authors · 2023-11-19
> **Response to Reviewer QTwu (Part 2/3)**
>
> **Question2:Datasets are limited. All of the datasets are about arithmetic questions. Some mentioned related studies like Auto-CoT (Zhang et al., 2022) are not compared in the experiments, especially while their studies are more general and conducted on more datasets.**
>
> **Response:**
> Thank you for your suggestions regarding our experiments. Based on your advice, we have added the Auto-CoT baseline and conducted supplementary experiments on the Q&A task, similar to the related experiments conducted with Auto-CoT and Complex-CoT baselines. The specific experimental results are as follows:
>
> Table 2: Supplementary experiments for the Auto-CoT baseline in text-davinci-003.
>
> | Model | GSM8K | AQUA |SVAMP|
> |  ----  | :----:  | :----:  | :----:  |
> | CoT     |55.1| 35.8 |77.3|
> | Complex-CoT   |66.8| 46.5 |78.3|
> | **Auto-CoT**   |60.7| 42.5 |80.0|
> | EPR     |64.6| 45.0 |84.6|
> |CEIL     |63.7| 47.2 |81.3|
> | DQ-LoRe    |69.1| 48.0 |85.0|
>
> Table 3: Supplementary experiments for the Auto-CoT baseline in gpt-3.5-turbo-16k.
>
> | Model | GSM8K | AQUA |SVAMP|
> |  ----  | :----:  | :----:  | :----:  |
> | CoT     |77.0| 51.9 |82.0|
> | Complex-CoT   |79.3| 57.0 |79.3|
> |  **Auto-CoT**   |78.4| 50.4 |86.0|
> | EPR     |77.3| 57.8 |88.0|
> |CEIL     |79.4| 54.7 |87.3|
> | DQ-LoRe(PCA)    |80.7| 59.8 |90.0|
> | DQ-LoRe(Gaussian kernel)    |79.0| 58.7 |88.3|
>
>
>
> Table 4: Supplementary experiments for the QA task in gpt-3.5-turbo-16k.
>
> | Model | StrategyQA | QASC |
> |  ----  | :----:  | :----:  |
> | CoT     |73.8| 81.8 |
> | Complex CoT     |74.5| 75.8 |
> |  **Auto-CoT**   |71.2| 74.1 |
> | EPR     |73.4| 80.2 |
> | CEIL    |73.4| 80.8 |
> | DQ-LoRe(PCA)    |74.6| 81.2 |
> | DQ-LoRe(Gaussian kernel)    |75.4| 82.7 |
>
> We have newly added experiments on the StrategyQA[1] and QASC[2] datasets, as well as a new baseline, AutoCoT[3].
>
> For the question-answering dataset, we introduce LoRe with Gaussian kernel, where the PCA step is replaced with a Gaussian kernel to compute the similarity between the Exemplar's embedding and the current inference question + CoT (obtained from the first query) embedding.
>
> The motivation for adopting the kernel method here is to preserve as much information as possible in the embeddings for Q&A tasks. Although this is not a dimensionality reduction approach, the fundamental idea remains unchanged. That is, first, the embeddings of $M$ similar Exemplars are queried, then mapped onto another vector space. After this mapping, the similarity of these embeddings to the current problem's question+CoT embedding is recalculated, followed by re-ranking to yield the final $N$ Exemplars ($M$>$N$).
>
>
> Based on these experiments, we find that,
>
> 1. Firstly, the performance of AutoCoT is generally inferior to retrieval-based methods such as EPR and CEIL, particularly in Q&A tasks, where its performance is even lower than CoT.
>
> 2. Secondly, in the new Q&A tasks, we are state-of-the-art, and an interesting discovery is that using a kernel trick to map the Exemplar's embedding into a higher-dimensional space for reordering, instead of PCA for dimensionality reduction, can effectively enhance the model's performance. There is even a 1.5% performance improvement on the QASC task compared to PCA.For Q&A tasks, there is a significant difference compared to solving math problems. Specifically, the background knowledge required for Math word problems is presented within the question itself, while the factual information for Q&A tasks needs to be queried from Large Language Models (LLMs). Moreover, this task is sensitive to textual details. Therefore, filtering information in embeddings is not a good choice for Q&A tasks, as dimensionality reduction could remove some vital entity information, which is not suitable for Q&A tasks. Hence, we have adopted a kernel method to transform the embeddings, ensuring that the information within the embeddings is not lost while still becoming separable.
>
>
> **Reference**
> >[1] Geva, M., Khashabi, D., Segal, E., Khot, T., Roth, D., & Berant, J. (2021). Did aristotle use a laptop? a question answering benchmark with implicit reasoning strategies. Transactions of the Association for Computational Linguistics, 9, 346-361.
>
> >[2] Khot, T., Clark, P., Guerquin, M., Jansen, P., & Sabharwal, A. (2020, April). Qasc: A dataset for question answering via sentence composition. In Proceedings of the AAAI Conference on Artificial Intelligence (Vol. 34, No. 05, pp. 8082-8090).
>
> >[3] Zhang, Z., Zhang, A., Li, M., & Smola, A. (2022). Automatic chain of thought prompting in large language models. arXiv preprint arXiv:2210.03493.
>
> >[4] Xu, S., Pang, L., Shen, H., Cheng, X., & Chua, T. S. (2023). Search-in-the-Chain: Towards the Accurate, Credible and Traceable Content Generation for Complex Knowledge-intensive Tasks. arXiv preprint arXiv:2304.14732.

---

> ### Author Response · Authors · 2023-11-19
> **Response to Reviewer QTwu (Part 3/3)**
>
> **Question3:I wonder if the authors conduct significance tests on the improvements of the proposed method over baseline methods**
>
> **Response:**
> We conducted new significance tests on our method. Due to the need for multiple inferences, this would entail substantial resource consumption. Our model underwent 10 repeated experiments on datasets such as GSM8K, AQUA, and SVAMP using gpt-3.5-turbo to calculate the mean and variance of our model, as well as the mean and variance of our Dual-Query baseline on the GSM8K dataset. The experimental results are as follows:
>
> Table 4: The average performance and variance of our model across various datasets.
>  | Statistic | GSM8K | SVAMP| AQUA|
> |  :----: | :----:  | :----:  |:----:  |
> | Average   |80.9| 88.8 |59.4|
> | Variance  |0.017| 0.889 |0.297|
>
> Table 5: The average performance and variance of our model compared to the baseline.
>  | Statistic| DQ-LoRe | DQ-LoRe w/o LoRe|
> |  :----: | :----:  | :----:  |
> | Average   |80.9| 79.0 |
> | Variance  |0.017| 0.043 |
>
> From Tables 4 and 5, we can observe that the fluctuation in the performance of the model is not very significant. It demonstrates the robustness and effectiveness of our method.

---

> ### Author Response · Authors · 2023-11-21
> **Waiting for further discussion.**
>
> Dear Reviewers,
>
> Thank you very much for your invaluable feedback on our paper. We have meticulously reviewed each of your points and endeavored to address them thoroughly. We would greatly appreciate it if you could review our responses and let us know if you have any additional comments regarding the paper or the rebuttal. We are eager to embrace all your critiques and integrate them into our work.
>
> Thank you!

---

> > ### Comment · Reviewer_QTwu · 2023-11-23
> >
> > I acknowledge that I have read all of the author responses.

---

> > > ### Author Response · Authors · 2023-11-23
> > >
> > > Thank you for your prompt response. If you have any additional questions or concerns, please feel free to share. Additionally, It would appreciate it if you could consider the possibility of a score adjustment.

---

> ### Author Response · Authors · 2023-11-23
>
> If our revised manuscript and rebuttal more closely meet your expectations for the paper, we respectfully ask you to reconsider your initial rating.
>
> If you have any further questions or require more information to raise your initial score, please feel free to let us know.

---

### Author Response · Authors · 2023-11-22
**General response to all reviewers**

Dear Reviewers and AC,

Thank you very much for the helpful reviews. We are thankful for the thorough suggestions on our previous manuscript. We have taken all the suggestions and made major changes to our previous draft. Our final version will be based on the rebuttal revision we newly submitted. We mainly have the following changes:


* Add Different Encoders: BERT-base-uncased and RoBERTa-base
* Add AutoCoT experiment result
* Add additional dataset: StrategyQA and QASC
* Analyze the latency of different methods
* Include more LLMs (such as LLama2 and Vicuna) to demonstrate the generalizability of the method
* Present experiments of DQ-Lore with other forms of regularization, such as MGS, Gaussian Kernel and so on.
* We extensively compared the performance of models without employing re-ranking algorithms and those using only CoT for retrieval.
* The effect of a trained encoder, pointing out the importance of training.
* We provided a detailed explanation of the distinctions between our work and others, as well as the novelty of our approach.

Thank you very much for your kind attention!

Best regards,

Authors

---

### Meta-Review · Area_Chair_nsR4 · 2023-12-31

**Metareview:**

The authors propose DQ-LoRe, a method for optimizing in-context learning (ICL) examples included in an LLM prompt based on two primary innovations: (1) a dual query to obtain 'reasoning' knowledge from the LLM (e.g., chaiin-of-thought) and (2) using a combination of the original query and the knowledge obtained from the previous query to retrieve exemplar(s) from a dimension-reduced space to include in the final prompt. Thus, the conceptual improvements are to improve ICL example selection with a combination of the CoT information and retrieval using a lower dimensional embedding (which admittedly makes intuitive sense). Experiments are conducted on three word problem datasets (in the original submission) and compared against multiple CoT-based systems -- showing improvements in most cases and competitive performance is all cases. Additionally, experiments were performed to assess performance under test distribution shift (i.e., covariate shift) based on additional word problem datasets (Matharith, SingleEq), an ablation study to evaluate the relative impact of the dimensionality reduction step and CoT-based retrieval, sensitivity with respect to initial exemplar for CoT retrieval, visualization, and a case study.

Consensus strengths identified by reviewers regarding this submission include:
- Including CoT in the ICL exemplar retrieval procedure makes sense in 'complex' problems where CoT has been shown effective as it will better bias the generation procedure and retrieve more relevant examples.
- The use of dimensionality reduction required additional thought and is convincingly shown to improve performance. Thus, this may influence other ICL selection-based work (assuming this continues to be an important paradigm).
- While this work is clearly influence by related methods (i.e., it is not a conceptual breakthrough), the novelty is sufficient while also being simple to understand and demonstrably effective. In this vein, the paper is also easy to understand overall.
- The empirical performance is generally a strong improvement  for both in-domain and domain-shifted settings.
- The secondary experiments do provide strong insight to the dynamics governing why DQ-LoRe works well.

Conversely, consensus limitations included:
- There were some questions regarding the specific methodological novelty. In my own reading, I thought it was relatively clear, but reviewers had continued clarification questions regarding specific claims of novelty. However, this can be handled in writing.
- Experiments are only conducted on math word problems (in the original submission). As other CoT works have expanded to other domains, it would be nice to see if DQ-LoRe works in other domains. Note that this was addressed in rebuttal and added to a new version of the manuscript for more general QA settings (with positive results).
- There were some questions regarding missing baselines and the use of additional LLMs, which was also addressed in the rebuttal -- and demonstrated positive results.
- The reviewers requested additional discussion regarding practical concerns (e.g., efficiency, complexity), which was also effectively addressed in the rebuttal (showing an increase, but still acceptable for many cases where CoT is already used).

Overall, this is an interesting work that took a conceptually interesting approach and worked through the technical details to get it to work for multiple domains and configurations at an acceptable cost in latency. I think that the original submission was borderline, but the additional results presented during rebuttal makes for a stronger paper that I believe would be sufficiently interesting to the broader community and potentially impactful. My only apprehension is that the new version of the paper (that incorporates new results provided during rebuttal) still requires a fair amount of polishing, but I believe this can be handled during preparation of a camera ready version.

**Justification For Why Not Higher Score:**

I believe the originally submitted work was borderline due to its focus on math word problems and, to a much lesser degree, inclusion of more LLMs, baseline comparisons, etc. However, the results added during rebuttal make this a much stronger submission. While the authors incorporated these into a new draft, I would like to see a polished version of this draft (including further discussion of the empirical results) before recommending a higher score.

**Justification For Why Not Lower Score:**

The work is original, the potential impact is sufficient, the method is convincingly shown to work well, and the paper is well-written.

---

### Decision · Program_Chairs · 2024-01-16

Accept (poster)